# LoRA: A Logical Reasoning Augmented Dataset for Visual Question Answering

**Jingying Gao**
School of Computer Science & Engineering
The University of New South Wales
jingying.gao@unsw.edu.au

**Qi Wu**
School of Computer & Mathematical Sciences
The University of Adelaide
qi.wu01@adelaide.edu.au

**Alan Blair**
School of Computer Science & Engineering
The University of New South Wales
a.blair@unsw.edu.au

**Maurice Pagnucco**
School of Computer Science & Engineering
The University of New South Wales
morri@cse.unsw.edu.au

## Abstract

The capacity to reason logically is a hallmark of human cognition. Humans excel at integrating multimodal information for logical reasoning, as exemplified by the Visual Question Answering (VQA) task, which is a challenging multimodal task. VQA tasks and large vision-and-language models aim to tackle reasoning problems, but the accuracy, consistency and integrity of the generated answers is hard to evaluate in the absence of a VQA dataset that can offer formal, comprehensive and systematic complex logical reasoning questions. To address this gap, we present LoRA, a novel Logical Reasoning Augmented VQA dataset that requires formal and complex description logic reasoning based on a food-and-kitchen knowledge base. Our main objective in creating LoRA is to enhance the complex and formal logical reasoning capabilities of VQA models, which are not adequately measured by existing VQA datasets. We devise strong and flexible programs to automatically generate 200,000 diverse description logic reasoning questions based on the SROIQ Description Logic, along with realistic kitchen scenes and ground truth answers. We fine-tune the latest transformer VQA models and evaluate the zero-shot performance of the state-of-the-art large vision-and-language models on LoRA. The results reveal that LoRA presents a unique challenge in logical reasoning, setting a systematic and comprehensive evaluation standard.[1][2]

## 1 Introduction

Logical reasoning is a fundamental hallmark of human cognition that enables us to understand and solve various problems, from everyday life to scientific research. Logical reasoning is the ability to analyze and identify logical relationships and derive conclusions or solutions based on known information or conditions [6, 3]. Humans are adept at integrating multimodal data for logical reasoning, such as visual, linguistic, or auditory information. A challenging task in artificial intelligence that requires such multimodal logical reasoning is Visual Question Answering (VQA)[1]. However, existing VQA datasets lack questions with complex logical structure, limiting the ability of these models to perform complex inference and multi-step reasoning. In this work, we introduce a new VQA dataset LoRA (Logical Reasoning Augmented Dataset), to address these challenges and

---

[1]LoRA Dataset Project page: https://lora-vqa.github.io/
[2]The LoRA Dataset code is available at: https://github.com/CarolineGao/LoRA-Dataset.git

37th Conference on Neural Information Processing Systems (NeurIPS 2023) Track on Datasets and Benchmarks.

explore the logical reasoning capabilities of VQA and large vision-and-language models, and how they perform as the logical difficulty increases.

State-of-the-art VQA models have made significant strides in tackling reasoning problems that hinge on visual relationships and basic logical constructs such as 'and', 'or', and 'not'. Nonetheless, they struggle to perform complex inference and multi-step reasoning [40, 36, 27, 11]. Large vision-and-language models can execute complex multi-step reasoning. However, even the most advanced models still often produce logical errors. They exhibit a tendency to fabricate facts when facing uncertain information [37, 14]. It is unclear to what extent they understand the task, are capable of reasoning logically and generalizing, even if they perform well on the benchmark.

This can be largely attributed to the limitations of existing VQA datasets[40, 23], which primarily offer questions with basic logical connections and not only fail to challenge the models to reason at higher levels of complexity, but also neglect to consider the logical difficulty and the performance of different approaches with increasing complexity. We studied and examined the logical syntactic complexity of the questions in several widely-used VQA datasets from 2016 to 2022 [1, 28, 20, 21, 35, 33, 25, 31, 9, 26] based on various logical categories and confirmed this limitation.

We introduce LoRA, a Logic Reasoning Augmented VQA Dataset, which addresses these issues with a focus on formal, complex, and diverse logical reasoning. The LoRA Dataset comprises 200,000 sophisticated and diverse logical reasoning questions based on the formal Description Logic SROIQ, based on realistic kitchen scenes, with ground truth answers, and logical prompt annotations. We devised multiple strong and flexible tools to automatically generate the LoRA dataset. The logical problems in LoRA span a wide and diverse range of complex logic, divided into three levels of increasing difficulty from simple to complex. This progression provides a more comprehensive evaluation of existing VQA methods, assessing to what extent their logical reasoning abilities have evolved, and highlighting for which logical reasoning tasks their performance is superior.

One example from our LoRA dataset is shown in Figure 1. These questions contain a number of different types of complex logical reasoning relationships, such as conjunction, disjunction, negation and "if ... then" rule-based reasoning, etc. In order to answer such questions involving multiple different logical relationships, multi-step logical reasoning is required, based on diverse types of logical inferences.

Our experiments show that current multimodal methods[19, 32, 24, 39, 13, 7, 38] fail to achieve satisfactory performance on the LoRA dataset and their performance decreases with the increasing difficulty level of the questions. The evaluation encompasses end-to-end training, transformer models within a fine-tuning setting, and large vision-and-language models in both zero-shot and few-shot learning settings on LoRA.

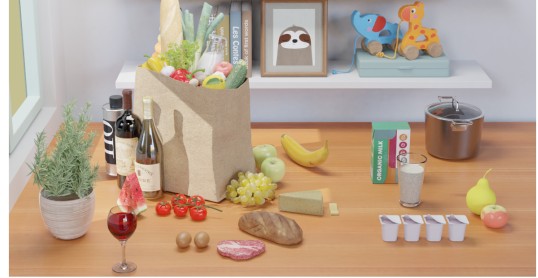

Q: If we do not have milk, is there another dairy product that does not necessarily contain fat but is rich in protein that can be substituted for breakfast? A: Yogurt
Q: Can we use the food between eggs and bread to make a meal for vegetarians? A: No

Figure 1: Sample image and questions from LoRA. Test questions are a combination of logical reasoning including rule-based conditional logic, conjunction, and negation.

In summary, our contributions are three-fold: (1) we present LoRA, a novel VQA dataset that challenges the state-of-the-art models to solve 200,000 multimodal logical reasoning questions of high complexity and diversity; (2) our dataset comes with automated scripts that enable researchers to create and enrich their own logical questions, facilitating the growth and variety of logical reasoning datasets; and, (3) we leverage a formal definition of logical difficulty based on Description Logic to systematically evaluate the logical reasoning abilities of existing VQA and large vision-and-language models across different levels of complexity, revealing important directions for future research.

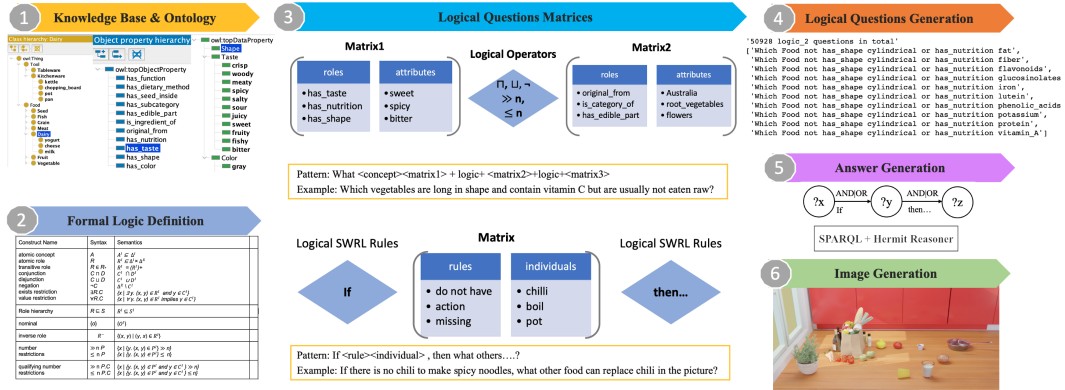

Figure 2: LoRA Dataset Construction Pipeline. Begins with a manually constructed knowledge base and formal description logic definition, then traverses the matrices to automatically generate questions, answers and images sequentially.

## 2 Related Work

We examined the logical complexity of questions in several widely-used VQA datasets from 2016 to 2022, including VQA-v2[1], GQA[20], OK-VQA[28], CLEVR[21], CLEVRER[35], and ScienceQA[26]. We conducted an assessment of the syntactic complexity of the questions, taking into account a variety of logical categories. These include the presence of distinct logical connectives (for instance, conjunction, disjunction, or negation), multiple appearances of the same logical operator within a single question, and mixtures of various logical operators [12]. Furthermore, we analyzed the questions that exhibit a rule-based logical framework, like "if-then" propositions, as well as questions that incorporate logical restrictions, including existential restriction, role hierarchy, and qualifying restriction, and so on.

Our research indicates that existing datasets are predominantly composed of questions that depend exclusively on visual-spatial information and do not require external logical reasoning, or they involve basic logical 'AND' or 'OR' connectives, as illustrated in Table 1. Therefore, to evaluate the capabilities of current VQA methods in tackling intricate logical questions, it is essential to possess a wide-ranging dataset that includes a substantial number and variety of such logical questions.

| Logic Categories | Logical Syntax | VQA-v2 | GQA | OK-VQA | CLEVR | CLEVRER | ScienceQA | LoRA |
|---|---|---|---|---|---|---|---|---|
| Conjunction | $C \sqcap D$ | yes | yes | yes | yes | yes | yes | yes |
| Disjunction | $C \sqcup D$ | no | yes | yes | yes | yes | yes | yes |
| Negation | $\neg C$ | yes | yes | yes | yes | yes | yes | yes |
| Compound of conjunction and disjunction | $C \sqcap D \sqcup E$ | no | no | no | no | no | no | yes |
| Compound of conjunction, disjunction and negation | $C \sqcap D \sqcup \neg E$ | no | no | no | no | no | no | yes |
| Existential Restriction | $\exists R.C$ | no | no | no | no | no | no | yes |
| Universal Restriction | $\forall R.C$ | no | no | no | no | no | no | yes |
| Inclusion | $R \sqsubseteq S$ | no | no | no | no | no | no | yes |
| Inverse Role | $R-$ | no | no | no | no | no | no | yes |
| Qualifying Number Restrictions | $\geq nP.C, \leq nP.C$ | no | no | no | no | no | no | yes |
| Concept Hierarchy | $C \sqsubseteq D$ | no | no | no | no | no | no | yes |

Table 1: Comparison of Logical Complexity between LoRA and Existing VQA Datasets.

## 3 The LoRA Dataset

The LoRA dataset is designed to evaluate and enhance the logical reasoning capabilities in VQA tasks. It includes a diverse range of questions with varying degrees of logical compositionality, spanning from simple to complex logical difficulties. These questions involve logical constructs such as conjunction, disjunction, negation, restriction, etc., random combinations of multiple logical operators, and conditional logic (e.g., if..then rule-based questions). To accurately answer questions in LoRA, complex logical reasoning is required, including multiple inference steps querying a knowledge base while leveraging visual information. LoRA comprises 200,000 diverse logic reasoning questions

based on the formal Description Logic SROIQ[2], along with ground truth answers, logical prompt annotations, and 100,000 realistic kitchen scenes.

The LoRA dataset is created automatically via multiple flexible automated scripts. Figure 2 provides an overview of the LoRA generation pipelines. The dataset creation process includes five steps: (1) constructing the ontology, (2) formal description logic definitions, (3) automatically generating questions, (4) dynamically queried answers, (5) automatically generating images.

In constructing the LoRA dataset, we first created a kitchen domain-specific ontology and defined its logic syntax based on Description Logic. We implemented a script to automatically generate complex reasoning questions, enabling arbitrary expansion of question quantity and complexity. Furthermore, we designed a dynamic query algorithm for the automatic retrieval of answers. For realistic visual context, we developed a script that can be executed within the Blender[5] environment to automatically generate realistic kitchen scene images corresponding to the questions. The ground truth answers are intentionally obfuscated by noise distractors to increase the level of difficulty. Last but not least, logical operators to construct the questions are included as logical prompt annotations. It should be emphasized that our dataset creation approach and pipeline are domain-agnostic, allowing adaptation to diverse knowledge areas.

## 3.1 The Ontology

We used OWLReady2[22] to create our initial ontology (Knowledge Base, KB), and drew inspiration from public food ontologies such as FoodOn [10] and FoodKG [15]. This ontology was framed using the industry-standard OWL format. The ontology defines domain knowledge through three elements: concepts (e.g., Food, Fruit), roles that represent atomic relations, (e.g., hasTaste); and individuals (e.g., apple). The ontology[2] is defined as KB = (A, T), where: KB = ABox + TBox.

The **ABox** contains specific instances of the TBox concepts (e.g., apple, banana) and their attributes, covering both visible (e.g., red) and invisible attributes (e.g., sweet). The **TBox** defines concepts (e.g., Food, Fruit, Vegetable, Meal), along with relationships between them(e.g., isSubClassOf, isMadeFrom). Additionally, Semantic Web Rule Language (SWRL)[18] rules are defined in the TBox for conditional logical reasoning.

The ontology provides formal representations and definitions for food and kitchen domain knowledge. The ontology provides three advantages: (1) it provides a rich vocabulary and knowledge base for question generation, (2) it standardizes and maps concepts to objects in images, (3) it enhances reasoning capabilities by enabling inference based on logical rules and knowledge. Our approach and pipeline to create the dataset is generalizable. It can work with any ontology (knowledge base) that adheres to the OWL specifications, which is the standard W3C Web Ontology Language.

## 3.2 Formal Description Logic Definition

Description Logic (DL)[2] is a family of formalisms for representing and reasoning with ontological knowledge. We use Description Logic syntax to formulate the logical questions in LoRA, as defined in Definition 1. Description Logic provides a standard way to construct the logical components of the questions, enabling diverse and sophisticated logical reasoning.

**Definition 1** (Description Logic Syntax). *Let C, D be concepts, R is a role (possibly inverse), P is a simple role (possibly inverse), n is a non-negative integer; then $C \sqcap D$, $C \sqcup D$ , $\neg C$, $\exists R.C$, $\forall R.C$, $\geq nP.C$, $\leq nP.C$, are also concepts (Table 2).*

| Description Logic Syntax | |
|---|---|
| Logic Categories | Syntax |
| atomic concept | C, D |
| atomic role | R |
| transitive role | $R \in R^+$ |
| conjunction | $C \sqcap D$ |
| disjunction | $C \sqcup D$ |
| negation | $\neg C$ |
| concept inclusion | $C \sqsubseteq D$ |
| existential restriction | $\exists R.C$ |
| universal restriction | $\forall R.C$ |
| number restrictions | $\geq nP.C$ or $\leq nP.C$ |

Table 2: Description Logic Grammar.

## 3.3 Automatically Generated Questions

We propose an algorithm that generates complex logical reasoning questions by unrolling the on-

tology into dataframes and populating a matrix with concepts, roles, attributes, logical operators, and formulas. The questions are formed by randomly placing logical operators in the matrix, ensuring diverse reasoning types and difficulty levels. The generated questions exhibit human-like logic, involving both multimodal information and commonsense knowledge as well as symbolic logic.

Logical questions are automatically generated using a five-step procedure: 1) unroll the ontology, 2) generate logical questions with logical operators, 3) generate conditional logical questions with rules, 4) filter rules to avoid repetition, and 5) enrich question diversity using synonymous phrases.

### 3.3.1 Unroll Ontology

We propose an algorithm that unrolls the ontology into a table format in three steps. First, we recursively extract the class hierarchy, relationship tuples, and object pairs from the ontology. For example, Vegetable is a subclass of Food, which is a subclass of Thing; (hasColor, Vegetable, Color) is a relationship tuple; and (carrot, orange) is an object pair. Second, we collapse the hierarchy into a list of dictionaries recursively, where each dictionary contains the entity and relationship information at each level. Finally, we transform the list of dictionaries into a dataframe with five columns: object classes, attribute classes, entities, attributes, and relationships. This dataframe is the basic building block for generating questions.

Our algorithm can unroll any ontology that is based on or can be converted to owlready2 [22, 17], not only our customized ontology. This is because the framework we utilized to construct the ontologies is the industry-standard OWL format. It can work with any other domain ontology (knowledge base) that adheres to the OWL specifications, which is the standard W3C Web Ontology Language [29].

### 3.3.2 Generate Logical Questions with Logical Operators

We propose an automated script to generate large numbers of logical questions with logical operators based on Description Logic (DL) syntax. We first define nine foundational logical operators based on DL syntax, which include AND, OR, NOT, inclusion, existential restriction, etc., shown in Table 3. These operators form a basic logical library that enables us to construct logical questions following the formal DL syntax.

To generate questions from an unrolled ontology represented as a table, we randomly select concepts, roles and entities to form a matrix of sentence blocks. We then apply logical operators to connect two or three sentence blocks within the matrix, resulting in logical questions with varying levels of complexity. Each row in the table corresponds to a branch of class, relation and entity pairs. By traversing the table, we can randomly combine class, relation and entity elements with diverse logical operators to create intricate questions with two or three layers of logical relationships.

The question template for the three-layer logical relationship is as follows: "$[QuestionTypes]$ Which | How many | Are there + $\langle concepts \rangle$ + $\langle$ sentence block 1 $\rangle$ + logical operator + $\langle$ sentence block 2 $\rangle$ + logical operator + $\langle$ sentence block 3 $\rangle$ ?" One such question could be: "$[Which]$ $\langle$ vegetables $\rangle$ in the picture exhibit $\langle$ a conical shape $\rangle$, **and** $\langle$ are rich in nutrient iron $\rangle$, **or** $\langle$ possess internal seeds $\rangle$?"

To illustrate, the logical operators used are not confined to [not], [and], [or], but also include formal description logic syntax defined above as a benchmark. Different logic syntax are inserted into the matrix, for instance, using $\geq nP.C$, we can generate a question like "which vegetables in the picture contain at least two different tastes, and are rich in cellulose?"; $C \sqsubseteq D$: class hierarchy: e.g. "Which foods in the picture are meat but can provide high protein and shaped solids?"; R- inverse role: "Which food in the image can be used to cook a meal for vegetarians?".

### 3.3.3 Generate Conditional Logical Questions with Rules

We extend our question generation pipeline to produce seven additional complex logical questions based on SWRL rules, illustrated in Table 4. For example, we use the following question template based on a SWRL rule: to cook [attribute] $\langle concept \rangle$, if we do not have $\langle object \rangle$, can we use other $\langle objects \rangle$ in the image instead? Question examples: to cook [spicy] $\langle noodles \rangle$, if we do not have $\langle pepper \rangle$, can we use other items in the image instead?

To generate complex logical questions, we need both the unrolled ontology table and the specific question templates. We design question templates based on SWRL rules and select and replace the

| Logical Reasoning Category | Logical Syntax | Logical Instance |
|---|---|---|
| Conjunction | $C \sqcap D$ | query(x): Fruit(x) $\sqcap$ hasTaste(x, juicy) $\sqcap$ hasNutrition(x, sweet) |
| Disjunction | $C \sqcup D$ | query(x): Fruit(x) $\sqcup$ hasTaste(x, juicy) $\sqcup$ hasNutrition(x, sweet) |
| Negation | $\neg C$ | query(x): Food(x) $\sqcap \neg$ hasTaste(x, spicy) |
| Existential Restriction | $\exists R.C$ | query(x): $\exists$ x (Vegetable(x) $\sqcap$ isIngredientOf(x, $y_1$) $\sqcap$ Dish($y_1$)) |
| Universal Restriction | $\forall R.C$ | query(x): $\forall$x(Fruit(x) $\rightarrow$ hasTaste(x, sweet)) |
| Role Hierarchy | $R \sqsubseteq S$ | Fruit(x) $\sqsubseteq$ Food(x), query(x): Fruit(x) $\sqcap \neg$hasTaste(x, juicy) |
| Inverse Role | $R-$ | query(y): Dish(y) $\sqcap$ hasIngredient(y, tomato) |
| Qualify Restriction | $\geq nP.C , \leq nP.C$ | query(x): $\exists$x(Fruit(x) $\sqcap$ hasTaste.min(x, 2, ActiveTaste)) |
| Concept Hierarchy | $C \sqsubseteq D$ | query($y_1$): $\forall y_1$.isClass($x_1$) $\sqcap \forall y_2$.isClass($x_2$) $\sqcap x_1 \sqsubseteq x_2$ |

Table 3: Basic Categories of Description Logic Reasoning Questions. Nine Types of Foundational Description Logical Reasoning.

concepts, roles and attributes from the table. According to the question templates, we traverse the table and generate complex logical questions corresponding to the SWRL rules.

For the Filter Rules and Enrich Questions sections, please refer to the supplementary material C.2 for more details.

### 3.4 Dynamically Queried Answers

A dynamic query algorithm was created to automatically retrieve answers from generated questions. SPARQL [30] was utilized to query answers for each individual clause in a question. The values of classes, relations, and attributes in each dataframe row (representing a question) were dynamically passed to the SPARQL variables as dynamic parameters for querying answers to each distinct clause. Then logical operations such as AND, OR, NOT, among others, between clauses, were handled by specialized algorithms. This methodology ensured the generation of ground truth answers. To avoid excessive analysis and ambiguity, constraints were set on the three-level logical relationship, and logical reasoning was processed following the language sequence.

### 3.5 Automatically Generated Images

We developed a Blender script to automatically generate realistic kitchen scene images, which include ground truth answers and random noise objects for added complexity. Specifically, LoRA images are rendered using Blender based on visible objects that include answers of each question and noisy objects. Each image in the LoRA dataset is associated with two questions.

We created four kitchen background scenes with different elements such as a table, a window, cupboards, shelf, plants, etc. Objects in the background such as fruits, vegetables, plants, tableware, etc., do not affect the reasoning of the answer. All the objects are consistent with the instances and attributes in the ontology, including 20 different vegetables, 20 different fruits, 4 meats, 3 fish, 4 dairy, etc. Visible objects include target answers and noisy objects, where noisy objects are randomly selected from various objects that do not include the actual answers. The script places the visible objects in random empty spots on the table in the given areas. This way, we can generate a rich and complex variety of images.

### 3.6 Logical Operator Annotations

In LoRA, logical annotations that indicate the logical operators used to construct logical questions are also provided. These annotations can be used as logical prompts to improve the performance of large language models on logical reasoning. We annotate each question with the corresponding logical operators used in this question.

## 4 Dataset Analysis and Comparison

The LoRA Dataset consists of 200,000 diverse description logic reasoning questions based on the formal Description Logic SROIQ over 100,000 realistic kitchen scene images. The questions require different degrees of logical compositionality and measure performance in logical difficulty from simple to complex. The answers in LoRA are either groups of objects, yes/no answers, or numbers

that require a combination of multi-modal information and multiple logical reasoning steps. Two different types of questions (such as how many, or which types, or if-then) share the same images. We have divided the dataset into three distinct subsets: 70% for training, 20% for validation, 10% for testing.

The LoRA dataset focuses on logical reasoning, offering a range of diverse and challenging question types. The domain of food and kitchen-related inquiries is utilized as a demonstrative example to showcase the diversity and complexity inherent in these logical questions. Furthermore, our generation scripts possess the flexibility to be applied across other domains.

## 4.1 Logical Reasoning Category Analysis

LoRA logical questions contain nine categories of basic description logic questions as shown in Table 3: conjunction, disjunction, negation, universal restriction, existential restriction, role hierarchy, inverse role, concept hierarchy, and qualifying number restrictions. These nine categories of Description Logic questions are the foundation of logical units.

The dataset also contains complex logical questions that have more diverse and complicated syntax. These questions are compound logical reasoning questions that combine two or three logical operators randomly selected from the nine basic categories. The first level of complex questions has two operators, while the second level has three operators.

The dataset has a balanced distribution of question types. The basic questions account for 20% of the dataset, the two-operator compound questions account for 30%, the three-operator compound questions account for 40%, and the conditional logical questions account for 10%. In particular, the LoRA dataset includes 4% SWRL rule-based conditional logical reasoning questions, and these can be divided into seven different categories as shown in Table 4.

| Logical Reasoning Category | Example | Answer | Inference Steps |
|---|---|---|---|
| Hypothesis replacement | To cook spicy noodles, if we do not have chilli, can we use any other items in the image instead? | spicy sauce | 6 |
| Functional comparison | To boil or fry eggs, which tools in the image can be used? | pot or pan | 3 |
| Similarity | To cook fried eggs with rice noodles, what alternative ingredients can replace rice noodles for a similar dish? | pasta | 5 |
| Avoiding | Which food in the image should be avoided when cooking a meal for vegetarians? | meat | 3 |
| Non-essential | Which foods in the image are not required to cook spicy food for vegetarians? | shrimp, salmon | 3 |
| Missing | What tool is missing to cook vegetable noodles? | pot | 3 |
| Dissimilarity | Which vegetables have at least two different nutrients compared to the one between apple and pear? | capsicum, onion, garlic | 5 |

Table 4: SWRL Rule-Based Conditional Logical Reasoning Questions.

## 4.2 Logical Reasoning Complexity Analysis

To objectively evaluate the complexity of logical questions, we formally define the concept of *question difficulty* [12], based on two criteria: the logical-syntactic complexity of the question and the complexity of the reasoning needed for the answer.

Firstly, the ***Question Syntax Complexity*** is defined by evaluating the quantity and diversity of logical connective operators used, with each additional connective or negation incrementally contributing to the complexity. Secondly, the ***Answer Inference Complexity*** is determined by the level of difficulty in deducing the correct answer through logical inference steps queried to a knowledge base. A more comprehensive elaboration of these definitions is provided in Section B of the supplementary material.

The logical problems in the LoRA dataset span a wide and diverse range of complex logic, divided into three levels of increasing difficulty from simple to complex. ***Level 1*** comprises basic Description Logic questions based on nine foundational operators, such as conjunction, disjunction, negation, restriction, and others. For example: "What are the names of the vegetables that are green in color in the picture and usually the leaves are the edible part?", ***Level 2*** includes questions that randomly combine two or three different types of logical operators, selected from the nine foundational operators in Level 1. For example: "Which foods in the picture are solid and do not necessarily contain fat but are rich in protein?", ***Level 3*** involves rule-based logical questions categorized into seven different categories based on Description Logic, where reasoning complexity increases with intricate rules, such as: "If I want to boil or fry eggs, which tools in the image can be used respectively?".

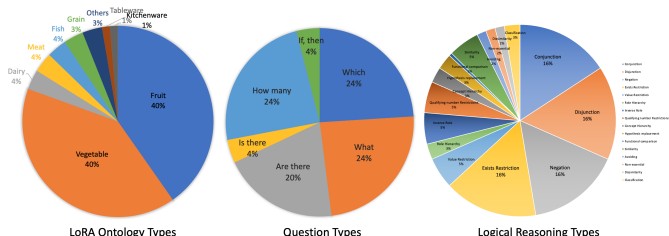

Figure 3: Dataset Statistics: Ontology types, question types, and logical reasoning types.

### 4.3 Question and Answer Types Analysis

**Question Types:** We categorize the question types in the LORA dataset into seven groups shown in Figure 3 based on the head of each question template: What, Which, How many, Is there ... , Are there ... , If ... then, and Why ... ? The distribution of each question type in the dataset is shown in Figure 3, which also presents the data distribution based on ontology types and the question distribution based on logical reasoning types. The lengths of the logical questions range from eight to thirty words.

**Answer Types:** The answers within the dataset fall into four distinct categories: yes/no responses; individual or multiple objects (e.g., pumpkin, cheese); numerical values; and a small proportion of specialized responses, including answers to 'why' questions that incorporate relations and attributes. Examples of such specialized responses include "because it has juicy taste". or instances where the answer is "None". The length of the answers ranges from two words to a maximum of 23 words, varying based on the answer type.

Supplementary material Section A offers a more comprehensive data analysis, encompassing the analysis of the ontology, question types, and the distribution of question logic in LoRA. Section A.3 of the supplementary material provides examples of question logical complexity and answer inference complexity in LoRA, ranging from 3 to 9 required inference steps.

## 5 Baseline Experiments and Analysis

### 5.1 Baseline Experiments

The VQA dataset was evaluated and analyzed via different VQA baseline models and large vision-and-language models. The evaluation results are shown in Table 5. In our methodology, we categorize the queries according to their intricacy, on a scale of 1 to 9. A score of 1 signifies a relatively straightforward query, while a score of 9 implies the necessity for more than nine logical deductions to infer the answer. In addition, a study of human performance was conducted, details of which can be found in Supplementary Section E.2.

**Ablation Study.** In order to carry out an ablation study and evaluate the capacity of baseline models to provide accurate answers without image data, question data, or external knowledge, we examined three models: a "deaf" CNN [4] model, a "blind" LSTM [16] model, and a CNN+LSTM model that made predictions based on visual and linguistic features, but without a knowledge base. The performance of all three models was found to be unsatisfactory, underscoring the importance of integrating visual, question, and knowledge base information for effective VQA reasoning.

**End-to-End Training Baseline.**

We trained MAC network [19] end-to-end on the LoRA dataset, inputting both visual and question language features to estimate the accuracy for each type of logical question. The MAC network struggled with rule-based and complex logical questions compared to simple ones, resulting in 64.6% accuracy.

**Fine-tuning Baseline.** Moreover, we trained and fine-tuned the VisualBERT model [24], based on the transformer architecture, on the LoRA dataset. Visual features were extracted using a pre-trained detectron2 model[34], while BERT[8] was used for question features. These image and language features were fed into vision-and-language fusion layers to predict final answers. Despite

the significant improvement in performance brought by the transformer of VisualBERT over other models, it displayed a decrease in its ability to answer complex questions as reasoning complexity increased, from 85.1% to 57.8%.

**Zero-shot Baselines.**

We evaluated the zero-shot performance of cutting-edge large vision-and-language models (VLMs), specifically, MiniGPT4 [39], Multimodal-GPT [13], InstructBLIP [7], and Multimodal Chain-of-Thought (MMCoT) [38] across three model types: in-context learning, instruct-tuning, and zero-shot CoT, using our LoRA dataset. We used the questions from the LoRA dataset as the prompts, with the accompanying images serving as the input images for these models for zero-shot experiments. Each question in the dataset not only guided the model in the direction of the answer but also implicitly highlighted the logical construct being evaluated. Our results revealed that all four models exhibited basic VQA capabilities, managing to answer questions that do not require much logical reasoning. They could also handle simple negation questions, but inconsistently. However, limitations were evident. The models occasionally ignore the visual information in the image and generate answers solely based on textual knowledge, especially MiniGPT4. Furthermore, they faltered in providing comprehensive responses involving multiple objects, often only listing one or two. Moreover, they are inconsistent and tend to fabricate answers that are often incongruent with the image content. Among the four models, InstructBLIP outperforms the other models with an average accuracy of 41.2%, but it performs poorly on complex logical questions that require more than six inference steps, achieving only 30.5% accuracy. MiniGPT4 struggles with negation questions that involve invisible attributes and complex logical questions involving multimodal inference. MMCoT exhibited suboptimal performance in VQA logical reasoning tasks, producing reasoning chains with logical inconsistencies.

**Few-shot Baseline.**

Furthermore, we evaluated Multimodal Chain of Thought based on the few-shot setting. We provided the logical operator that constructed each question as the prompting context to the model when asking it to answer each question. Using the logical operators of the questions as prompts improved the performance by 5.87% compared to not using them. Further analysis shows that logical prompts can help multimodal models stimulate logical thinking better.

## 5.2 Error Analysis

The following analysis investigates the different behaviors and performance of VQA models and multimodal LLM models, delving deeper into the underlying reasons behind the observed model behavior, and elucidates the potential factors that contribute to the strengths or weaknesses of a particular model.

### 5.2.1 MAC and VisualBERT's Challenges in Complex Logical Reasoning

The MAC network and VisualBERT model, despite their groundbreaking advancements in VQA, primarily rely on deep neural networks to derive implicit representations from the provided data. Deep learning models, by their very nature, often lack the inherent capacity to reason in an explicit, step-by-step manner that might be required for intricate logical constructs. Instead, these models tend to predict answers based on the patterns and associations they have learned. While this approach works efficiently for numerous tasks, it may not provide the rigorous, step-by-step logical reasoning required for our dataset, especially when questions increase in complexity.

### 5.2.2 Multimodal LLMs Baselines' Challenges in Complex Logical Reasoning

Building on our observations from the zero-shot and few-shot experiments, we further analyzed challenges faced by large vision-and-language models (VLMs). We observed that large vision-and-language models demonstrate deficiencies in handling logical constructs, especially with operators like negation and multiple logical operators appearing in a question. For example, when presented with negation-centric queries, these models tend to produce keyword-driven positive responses, overlooking inherent logical relationships. Such tendencies underscore that, despite expansive training data, these models lack genuine "logical understanding".

| Complexity of Inference Steps | Logical Operators | CNN | LSTM | CNN+LSTM | MAC | VBert | MGPT4 | MMGPT | InstBLIP | MCoT | Humans |
|---|---|---|---|---|---|---|---|---|---|---|---|
| 1 | C, D | 46.3 | 45.2 | 65.3 | 76.2 | 85.1 | 47.9 | 45.7 | 48.2 | 30.2 | 88.2 |
| 1 | R | 42.2 | 33.4 | 64.1 | 69.7 | 83.2 | 41.3 | 40.9 | 45.7 | 33.2 | 77.6 |
| 2 | $\neg C, \neg D$ | 43.0 | 31.8 | 62.1 | 72.6 | 80.6 | 40.7 | 39.5 | 44.5 | 29.6 | 78.4 |
| 2 | $\neg R$ | 35.78 | 39.65 | 61.3 | 70.5 | 80.2 | 34.5 | 40.2 | 43.6 | 28.4 | 76.5 |
| 2 | $C \sqsubseteq D$ | 35.64 | 38.25 | 60.2 | 68.4 | 78.3 | 43.2 | 41.0 | 45.4 | 29.7 | 72.9 |
| 3 | $C \sqcap D$ 
 $C \sqcup D$ 
 $D \in \{D, P \geq n, P \leq n\}$ | 36.8 | 38.2 | 61.3 | 69.2 | 80.2 | 45.7 | 44.2 | 45.6 | 32.5 | 82.2 |
| 4 | $\neg C \sqcap D$ 
 $\neg C \sqcup D$ 
 $D \in \{D, P \geq n, P \leq n\}$ | 36.2 | 35.6 | 60.2 | 67.8 | 75.3 | 41.8 | 38.5 | 42.9 | 26.5 | 75.6 |
| 4 | $C \sqsubseteq D, R_1 \sqcap R_2$ 
 $C \sqsubseteq D, R_1 \sqcup R_2$ | 31.3 | 30.8 | 59.6 | 65.2 | 74.3 | 41.2 | 42.5 | 43.9 | 27.6 | 78.2 |
| 5 | $C \sqsubseteq D, \neg R_1 \sqcap R_2$ 
 $C \sqsubseteq D, \neg R_1 \sqcup R_2$ 
 $\neg C \sqsubseteq D, R_1 \sqcup R_2$ | 28.7 | 29.3 | 53.2 | 59.7 | 68.8 | 40.1 | 39.7 | 41.7 | 26.9 | 70.6 |
| 4-5 | $C \sqcap D \sqcap E$ 
 $C \sqcup D \sqcup E$ 
 $C \sqcap D \sqcup E$ | 31.2 | 30.6 | 56.7 | 62.3 | 75.4 | 40.2 | 40.7 | 41.5 | 27.5 | 73.5 |
| 5-6 | $\neg C \sqcap D \sqcap E$ 
 $\neg C \sqcup D \sqcup E$ 
 $\neg C \sqcap D \sqcup E$ | 29.3 | 28.5 | 50.2 | 58.6 | 62.8 | 32.5 | 30.9 | 35.7 | 26.3 | 72.8 |
| 6 | $C \sqsubseteq D, R_1 \sqcap R_2 \sqcap R_3$ 
 $C \sqsubseteq D, R_1 \sqcup R_2 \sqcup R_3$ | 29.7 | 28.6 | 51.3 | 57.6 | 68.5 | 31.5 | 30.2 | 34.4 | 22.6 | 75.6 |
| 6-7 | $C \sqsubseteq D, \neg R_1 \sqcap R_2 \sqcap R_3 \geq n$ 
 $C \sqsubseteq D, \neg R_1 \sqcup R_2 \sqcup R_3 \leq n$ 
 $C \sqsubseteq D, \neg R_1 \sqcap R_2 \sqcup R_3 \leq n$ | 22.6 | 23.9 | 42.3 | 54.5 | 61.8 | 29.7 | 29.7 | 30.5 | 20.3 | 65.8 |
| $complexity(\varphi) + 2$ | if ..., then 
 $C, R \rightarrow D$ | 29.8 | 28.7 | 48.6 | 52.3 | 57.8 | 30.6 | 30.2 | 32.5 | 21.6 | 83.6 |
| | Learning | train set | train set | train set | train set | train set | zero-shot | zero-shot | zero-shot | few-shot | humans |

Table 5: Evaluation of Baseline Models over Varying Levels of Logical Complexity, based on Inference Steps and Question Syntax Complexity (Accuracy in %). Model names: VBert = VisualBert, MGPT4 = MiniGPT4, MMGPT = Multimodal-GPT, InstBLIP = InstructBLIP, MCoT = Multimodal CoT.

Furthermore, large vision-and-language models often ignore visual cues, leaning heavily towards text-driven answers. Given their significant training on textual data, the VLMs might inherently prioritize textual patterns over visual cues when formulating responses.

# 6 Conclusion

In this paper, we propose LoRA, a novel VQA dataset that challenges the complex and formal logical reasoning abilities of VQA models and large vision-and-language models. LoRA consists of 200,000 multimodal complex logical problems. It covers a wide and diverse range of logical reasoning types, divided into three levels of increasing difficulty. We have also provided automated scripts for generating logical questions and images, enabling researchers to extend and customize the dataset. We have conducted a comprehensive and systematic evaluation of several state-of-the-art VQA models and large vision-and-language models on LoRA, revealing their strengths and weaknesses in different logical reasoning tasks. Our results show that there is still a large gap between the current models to solve multimodal logical reasoning problems, especially for complex logical reasoning questions. We hope that LoRA will serve as a valuable benchmark and a source of inspiration for future research in multimodal logical reasoning.

# Acknowledgments

We would like to thank the anonymous reviewers for their valuable comments and suggestions. Additionally, our thanks go to Zhaofeng Yuan, Ross Zarghami, Lihan Li, Ben Brown, and Chris Ell for helpful discussions and suggestions. We also appreciate the support from the Blender Community, with special thanks to James Hurlock, Anna Trofimova, Chris Lee and John Joestar for assisting us with Blender.

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
