# A  Data Analysis

The LoRA Dataset Project page: https://lora-vqa.github.io/.

The LoRA Dataset Generation code is available at GitHub: LoRA Dataset Generation GitHub link. Link: https://github.com/CarolineGao/LoRA-Dataset.git

The LoRA Dataset can be found on GoogleDrive: LoRA Dataset.

Link:https://drive.google.com/drive/folders/1H4msZ4ae1msEDApxOT7cr1A1QwQ33Pkn

## A.1  The Ontology Analysis

The LoRA Ontology (Knowledge Base) consists of 12 categories of things, such as food, vegetables, fruits, meat, fish, grains, dairy products, seeds, tools, kitchen utensils, tableware, and plants. Each category includes around 20 classes, such as Apple, Banana, Orange. Each class includes different instances, such as red apple, green apple, etc. It also contains 15 relations and 101 attributes that describe the features and parameters of the categories and instances, such as Color, Taste, Country, Recipe, Edible, Dietary, etc. It contains 100 instance types. The categories and instances have a clear hierarchical relationship. For example, every vegetable, fruit, and meat sub-category belongs to the food category; food and utensils belong to things; and all plants belong to things. The ontology relationship is built based on formal Description Logic to define the relationships between classes, instances and attributes.

Figure 1 shows the ontology analysis that we performed to build our ontology. The ontology consists of different categories of concepts, instances and relationships that represent the knowledge in our domain. The figure also shows the proportion of each concept in the ontology by the size of the nodes. The larger the node, the more data it contains in that category.

## A.2  Question Statistics

### A.2.1  Question Word Frequency

Figure 2 (a) shows a word cloud of the most frequently appearing words in the question texts. Repeated words that do not contain any semantic meaning, such as "in", "the", "image", are removed to give us a clearer view of the semantic range of LoRA dataset. The diagram indicates that LoRA covers the main logic connectives, as well as a wide range of topics related to food, such as different categories of food, visible and invisible attributes and relationship information. The words from different topics are distributed across the word cloud.

Figures 2 (b) (c) (d) show the word clouds for each of the three levels of logical questions. The word clouds reflect the different levels of logical complexity in the questions. In Level 1 logical questions, words such as "not" and "and" appear frequently, indicating that our logical questions involve not only simple conjunctions but also negations. In Level 2 logical questions, words such as multiple logical operators are common, indicating that our logical questions involve comparisons and combinations of multiple logic connectives. In Level 3 logical questions, words such as "If we do **not**" are frequent, indicating that our logical questions involve conditional rules.

### A.2.2  Question Types Distribution

Figure 3 presents the distribution of question types based on categories of the question domain (Figure 3a), such as questions requiring knowledge of vegetable and fruit, which account for the majority, but also cover a wide range of the main food and kitchen domains. Additionally, question types are distributed according to queries (Figure 3b), such as questions beginning with "Which" or "What", verification type questions like "Is there" or "Are there", and conditional type questions such as "If...then".

Figure 4 illustrates the question length distribution for each level of logical questions. The word length for the three levels of logical questions varies from 8 to 30, indicating a higher compositional diversity compared to other VQA datasets. On the other hand, the word length of questions in other VQA datasets, such as VQA, GQA, and CLEVR, lies between 3 and 15. This indicates that our questions present more significant language understanding challenges as they do not solely consist

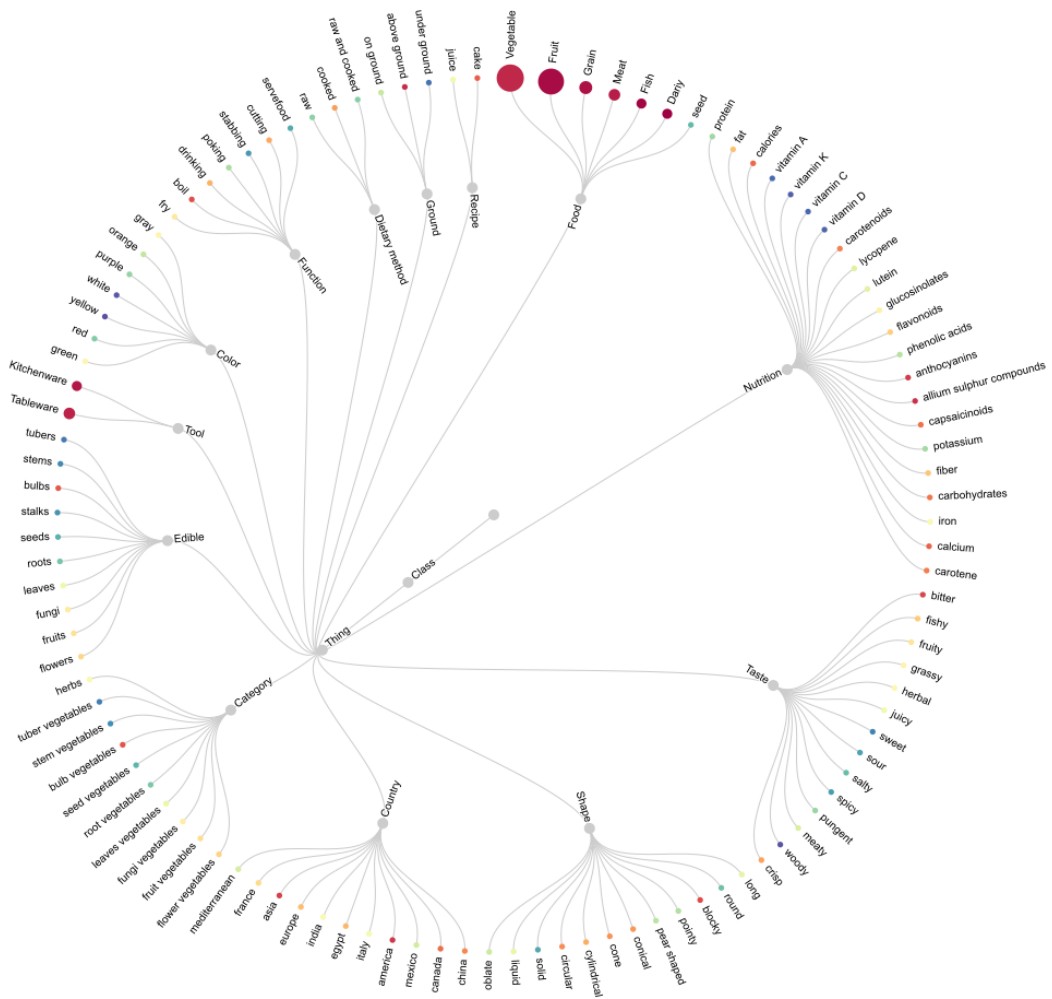

Figure 1: Ontology hierarchy analysis: concepts, relationships, and instances

of single queries. Instead, they involve a combination of multiple logical reasoning tasks, queries, comparisons, etc.

## A.3 The Distribution of Logics

The LoRA dataset offers a diverse array of logical questions, encompassing more than 16 different types. Answering these multimodal logical questions requires complex logical reasoning, requiring 3 to 9 steps to infer the correct response.

Table 1 on page 6 categorizes these logical questions based on the complexity of the inference steps and question syntax. It also provides examples and required information to infer the answers for each logical question category.

Figure 5 illustrates the distribution of logical reasoning question types, which spans a broad range. This includes conjunction, disjunction, existential restriction, universal restriction, inclusion, exclusion, conditional reasoning, and other complex logical questions. It's important to note that the conjunction questions in LoRA are not simple queries based on visual features; they demand much more intricate reasoning, which involves visual, linguistic, and knowledge-based logical reasoning.

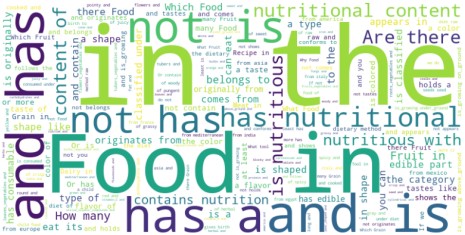

(a) Word cloud distribution in LoRA.

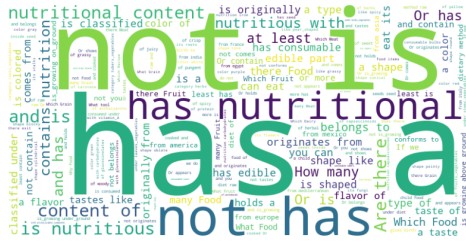

(b) Word cloud distribution for Level 1 logical questions.

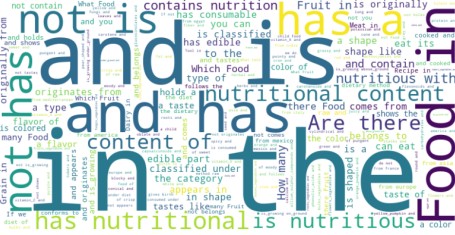

(c) Word cloud distribution for Level 2 logical questions.

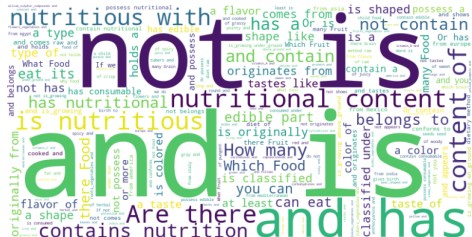

(d) Word cloud distribution for Level 3 logical questions.

Figure 2: Question word frequency

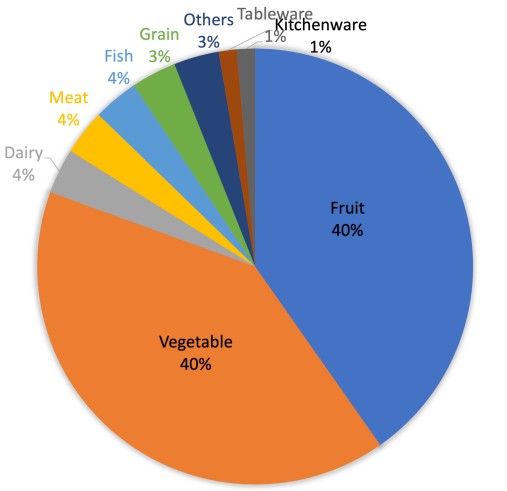

(a) Question types distribution by ontology categories

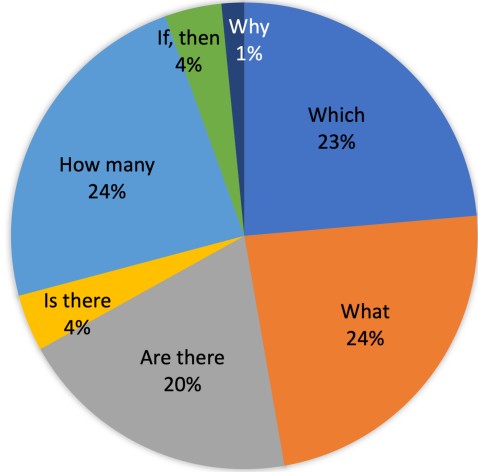

(b) Question types distribution

Figure 3: Question types distribution

# B  Question Difficulty

To evaluate how complex the logical questions are, we formally define the concept of *question difficulty*, based on two criteria: the logical-syntactic complexity of the question and the complexity of the reasoning needed for the answer.

## B.1  Question Logical Syntactic Complexity

We define question syntactic complexity by evaluating the quantity and diversity of logical connective operators used. The complexity is determined by the count of logical connectives, with each additional connective or negation incrementally augmenting the complexity. For rule-based questions, such as

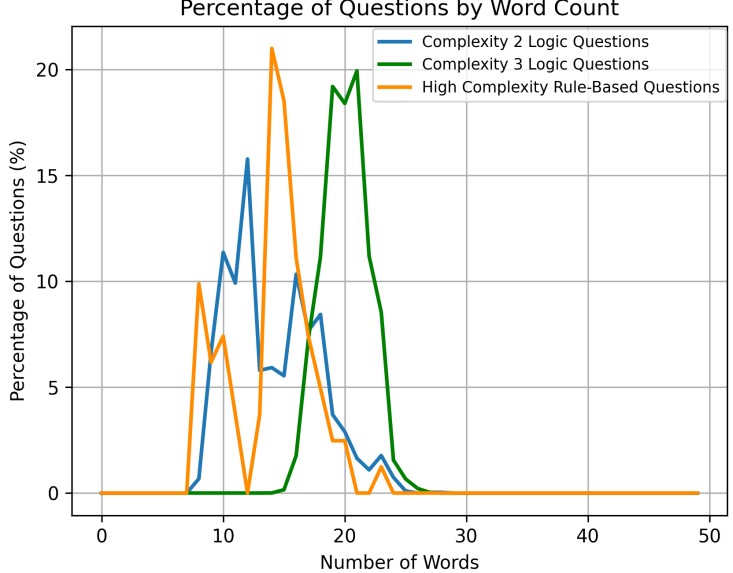

Figure 4: Question length distribution for each level of logical questions

"if... then" propositions, it can be considered as logical implications $C \rightarrow D \equiv \neg C \sqcup D$, the question complexity increases by 2 according to the logical syntax.

**Definition 1 (Logical Syntactic Complexity)** *Let D be the VQA dataset and K be the Knowledge Base. For each type of question in D, let $\varphi$ be a formula in Description Logic representing that question. We define the* syntactic complexity *of $\varphi$ as follows:*

*complexity($\varphi$) = {*

    *0,*

        *if $\varphi$ is a concept or role,*

    *complexity($\varphi_1$) + complexity($\varphi_2$) + 1 ,*

        *if $\varphi = \varphi_1 \, O \, \varphi_2$ , where $O \in \{ \sqcap, \sqcup, \geq, \leq \}$,*

    *complexity($\varphi_1$) + 1 ,*

        *if $\varphi = \neg\varphi_1$,*

    *complexity($\varphi_1$) + complexity($\varphi_2$) + 2 ,*

        *if $\varphi = \varphi_1 \rightarrow \varphi_2$ }*

### B.2 Answer Inference Complexity

The process and steps required to infer an answer from a seemingly simple question may actually be far more complex than finding an answer from a question with a complex grammatical structure.

Take for instance the query: "If we ran out of milk, are there other dairy food in the image that could be used as substitutes?" While this question is formulated as a straightforward, rule-based query, the inference process required to resolve it may be more intricate than what is needed for questions with more complex grammatical structures or for compound questions that employ multiple conjunctions, such as "Can you enumerate the quantities of red apples, yellow bananas, and tomatoes present in the image?" The complexity of reasoning and answering a question is not always directly proportional to the apparent complexity of the question's syntax.

The answer inference complexity is defined as the level of difficulty in deriving the correct answer through logical inference steps queried to a knowledge base. For compound connectives, we use a matrix to assign numerical values based on knowledge base queries, summing these to determine the

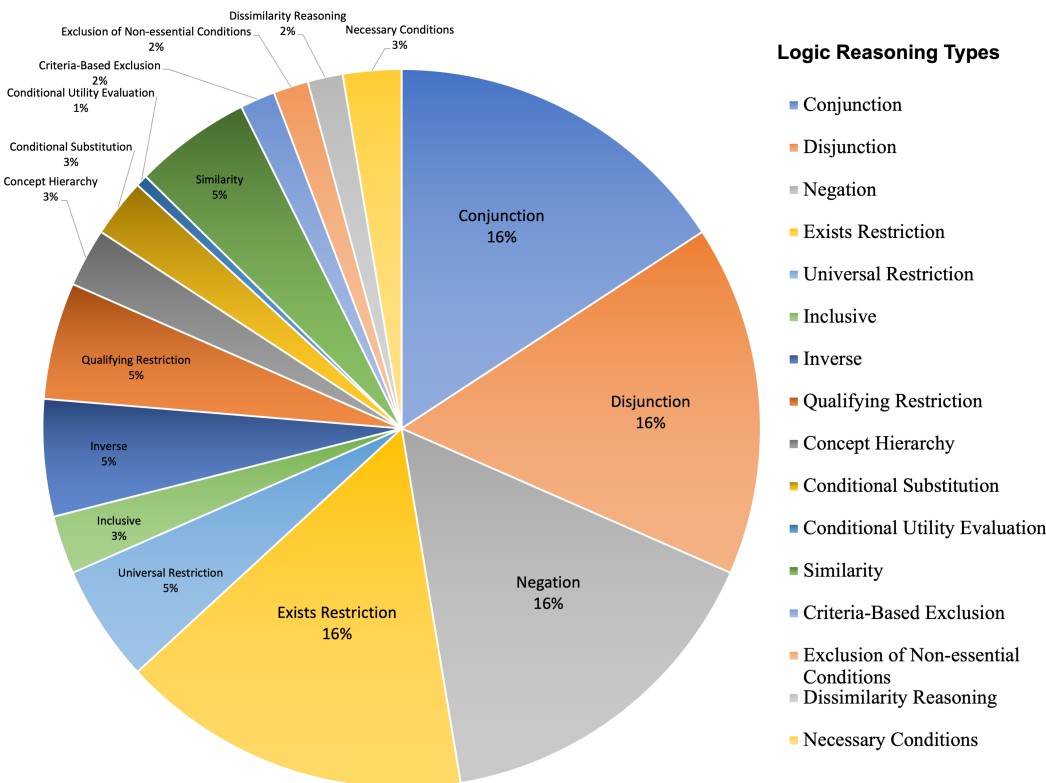

Figure 5: Logical reasoning question types distribution

necessary inference steps for answer derivation, attributing one point per query that cannot be simplified. For conditional reasoning, which involves drawing conclusions from "if-then" propositions, the inference steps are calculated by the reasoner checking each query in the knowledge base based on deductive reasoning.

## C  Dataset Generation Additional Details

This is additional content for Section 3 in the main paper. Our automation method could facilitate customization for dataset and extension to other domain knowledge areas.

### C.1  Ontology Creation Additional Details

We employed Owlready2 for the initial ontology framework construction. Owlready2 is a Python package for ontology-oriented programming, capable of loading, modifying, and saving OWL ontologies and performing reasoning using the embedded HermiT. OWL, the acronym for the W3C Web Ontology Language, is the industry-standard ontology format.

Our ontology defines domain knowledge through three elements: concepts (e.g., Food, Fruit, Vegetables, Tool), roles representing atomic relations (e.g., hasTaste, hasColor), and individuals (e.g., apple, pumpkin). Utilizing the OWL ontology structure, we populated the framework with 12 categories under 'Things' such as Food and Tool, 100 instance types, such as orange, carrot, egg, 101 attributes (e.g., Color: 'green', 'red'; Taste: 'bitter', 'sweet'), and 15 relations, such as hasTaste, hasColor. Further ontology analysis is detailed in Section A.1 The Ontology Analysis.

Post content integration, we designated relationships and attributes to individuals, linking them with concepts. An illustrative example is: `apple = Fruit("apple", has_color=[green, red], has_taste=[sweet], has_shape=[round], is_ingredient_of=[juice, pie])`. In addition, we also employed OWL Semantic Web Rule Language (SWRL) to frame rules, including conditional "if...then" statements.

| Complexity of Inference Steps | Logical Types | Question Type | Features | Logical Operator | Examples |
|---|---|---|---|---|---|
| 3 | $C \sqcap D$ | Query | V,Q,KB | AND | Which food has a taste similar to onion AND can used to make broth? |
| 3 | $C \sqcup D$ | Count | V,Q,KB | OR | How many vegetables in the picture are green in color OR usually have leaves as the edible part? |
| 3 | $\geq nP.C$ | Compare | V,Q,KB | Value Restriction | Is the quantity of the item to the right of the watermelon greater than the number of distinct fruits in the picture? |
| 3 | $\leq nP.C$ | Compare | V,Q,KB | Restriction,OR | Are there 10 or fewer eggs in the carton? |
| 4 | $\neg C \sqcap D$ | Verify | V,Q,KB | NOT, AND | Is the item to the left of the meat not suitable for vegans and is free of fat? |
| 4 | $\neg C \sqcup D$ | Query | V,Q,KB | NOT, OR | Are there any vegetables in the picture that are typically eaten cooked or are not categorized as stem vegetables? |
| 4-5 | $C \sqcap D \sqcap E$ | Query | V,Q,KB | AND, AND | Which food in the picture is a type of fruit and the same color as the avocado and has a pungent flavor? |
| 4-5 | $C \sqcup D \sqcup E$ | Query | V,Q,KB | OR, OR | Which food in the picture is a root vegetable that is orange in color or commonly used for Halloween decorations?? |
| 4-5 | $C \sqcap D \sqcup E$ | Verify | V,Q,KB | AND, OR | In the picture, is there a kitchenware item that is used for stirring, and is made of either wood or silicone? |
| 6-7 | $\neg C \sqcap D \sqcap E$ $\neg C \sqcup D \sqcup E$ $\neg C \sqcap D \sqcup E$ | Query | V,Q,KB | NOT,AND,AND | Which food in the picture is the same colour as the item directly behind the pumpkin but is not the closest item to the garlic, and is in the same half of the picture as the watermelon? |
| 4 | $C \sqsubseteq D, R_1 \sqcap R_2$ $C \sqsubseteq D, R_1 \sqcup R_2$ | Query | V,Q,KB | Inclusive, AND | Which green vegetable on the table contains nutrition with vitamin A, B and has consumable roots? |
| 5 | $C \sqsubseteq D, \neg R_1 \sqcap R_2$ $C \sqsubseteq D, \neg R_1 \sqcup R_2$ $\neg C \sqsubseteq D, R_1 \sqcup R_2$ | Query | VQKB | NOT, AND, OR | Identify the food on the table in the image that isn't classified as a fruit, but can be used to make juice or eaten raw? |
| 6-7 | $C \sqsubseteq D, \neg R_1 \sqcap R_2 \sqcap R_3$ $C \sqsubseteq D, \neg R_1 \sqcup R_2 \sqcup R_3$ $C \sqsubseteq D, \neg R_1 \sqcap R_2 \sqcup R_3$ | Query | V,Q,KB | NOT,NOT,AND,Compare | Which food in the picture is a type of vegetable that is not red inside, but is bigger than the item next to the avocado which is not a fruit, and has a flavour that is not sour? |
| 6-7 | $C \sqsubseteq D, \neg R_1 \sqcap R_2 \sqcap R_3 \geq n$ $C \sqsubseteq D, \neg R_1 \sqcup R_2 \sqcup R_3 \leq n$ $C \sqsubseteq D, \neg R_1 \sqcap R_2 \sqcup R_3 \leq n$ | Verify | V,Q,KB | NOT,AND,AND,Inclusive | Is there a dairy food in the picture that is not the same colour as the wine in the glass, but contains protein and can be used to make pasta sauce? |
| 3-4 | $C \rightarrow D$, if...then | Query | V,Q,KB | Necessary Conditions | What tool is missing for cooking vegetable noodles? |
| 4-5 | $C \rightarrow D$, if...then | Query | V,Q,KB | Conditional Utility Evaluation | I want to boil an egg or fry an egg, what tools in the image are suitable respectively? |
| 5-6 | $C \rightarrow D$, if...then | Query | V,Q,KB | Conditional substitution | If we don't have chili, what can we use as a substitute from the items in the image to prepare spicy noodles? |
| 5-6 | $C \rightarrow D$, if...then | Query | V,Q,KB | Alternative Item Substitution | If we want to cook fried eggs with rice noodles but don't have any rice noodles, what other ingredients from the image can we use to make a similar dish? |
| 6-7 | $C \rightarrow D$, if...then | Query | V,Q,KB | Criteria-Based Exclusion | Which foods in the image should be avoided when preparing a meal for someone who follows a vegetarian diet? |
| 6-7 | $C \rightarrow D$, if...then | Query | V,Q,KB | Exclusion of Non-essential Conditions | Which ingredients in the image are not necessary for preparing a spicy vegetarian dish? |
| 7-9 | $C \rightarrow D$, if...then | Query | V,Q,KB | Dissimilarity Reasoning | Which vegetables in the image have at least two nutritional differences compared to the vegetable located between the apple and pear? |

Table 1: Logical question types in LoRA with their complexity, syntax, semantics and examples. For each type of logical question, we provide the number of logical inference steps, the logical syntax, the question semantic type, and an example. V, Q, KB stand for visual, question and knowledge base information required to infer the answers.

Furthermore, Owlready's transparent access to OWL ontologies allows integration of any ontology complying with OWL specifications. To enhance and standardize our ontology's domain knowledge, we assimilated content from public ontologies like FoodOn and FoodKG, augmenting attributes and other domain-specific details.

## C.2 Question Generation Additional Details

The ontology, constructed on the standard OWL framework, comprises concepts, roles, and individuals. This structure enabled the design of an algorithm to transform the ontology into table dataframes. Refer to Section 3.3.1, Unroll Ontology in our main paper, for an in-depth discussion on the unrolling process. It should be noted that our method is capable of unrolling any ontology compatible with or convertible to owlready2, beyond just our custom-built ontology.

This dataframe forms the foundation for question generation. Our objective is to generate logical questions with distinct complexity levels. The methodology, rooted in populating matrices with elements from our ontology and formal description logics, involves strategically placing logical operators to ensure a variety of reasoning types and difficulty tiers. Details on the generation of foundational and Level 2 logical questions can be found in the main paper's Section 3.3.2 and more complex conditional logical question generation is detailed in Section 3.3.3, respectively.

This section supplements two subsequent steps of our question generation pipeline: (4) Filter Rules to Avoid Repetition and (5) Enrich Question Diversity.

### C.2.1 Filter Rules to Avoid Repetition

To avoid generating repetitive questions or questions without practical meaning, the below rules are set in the question generation engine: Rule 1: unique concat program to ensure the uniqueness of the question and no duplicate questions are generated; Rule 2: avoid conjunction or disjunction of two identical clauses; Rule 3: filter to maintain answer diversity by avoiding too many questions with the same answer; Rule 4: the generated question is limited to a compound sentence that uses two logical symbols to connect three sub-clauses to match the logic expression in human language.

### C.2.2 Enrich Question Diversity

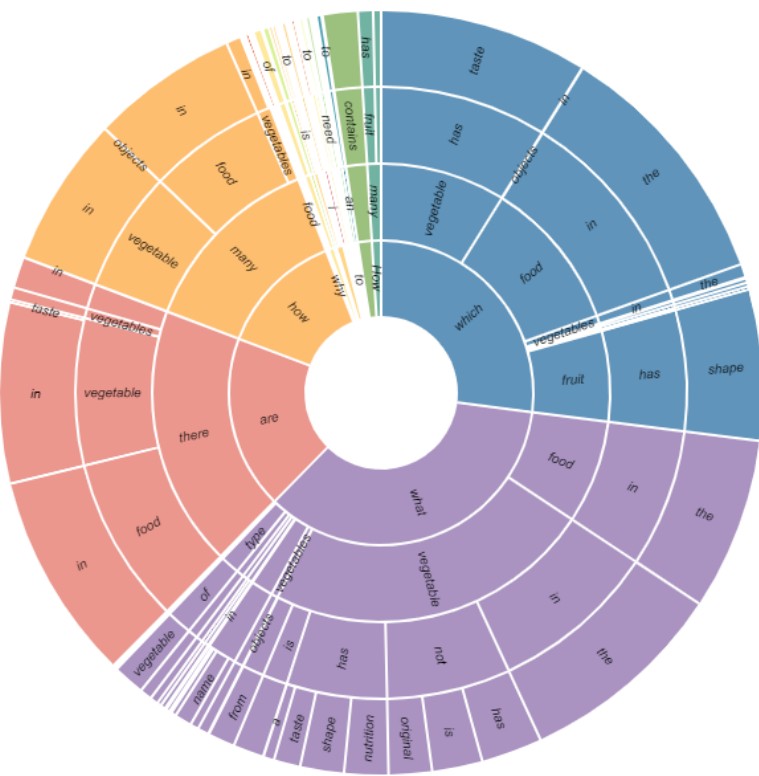

Figure 6: Distribution of LoRA questions by first four words

Figure 6 shows the distribution of LoRA dataset questions by first four words. It demonstrates that questions in LoRA dataset have a rich variety of types and range widely.

We develop a relation map program that uses a dictionary to match the same relation paradigm with various natural language expressions. This program replaces the original question expression and makes the regenerated question more natural and diverse. It also conforms to the expression and rich expressiveness of natural human language.

relation map = { 'has_for_taste': ['has the taste', 'tastes', 'tastes like', 'is'], 'has_child_food': [ 'is the parent of', 'contains the subcategory of', 'is made with the food'], 'apple': ['gala', 'fuji', 'pink lady'] }

In this way, the question language of our automatically generated template becomes varied and fluent. For example, "Which vegetables in the picture are commonly carved into decorative lanterns called jack-o'-lanterns for the Halloween season and are orange?"

### C.3   Image Generation Additional Details

For LoRA's image generation, we utilized Blender, an open-source platform, known for its comprehensive repository of realistic objects across various domains, including food and kitchen. With Blender's scripting capability (Python), we developed automated scripts to position objects in a scene, creating realistic images.

Our process ensures the uniqueness of each question-answer-image triplet. Upon crafting questions and their corresponding answers, we generate images reflecting the correct answers, augmented with random 'noisy' objects which are not in the correct answer range. The process initiates with setting a background in the scene, such as including a table, a window, and a shelf. All ontology-based individual objects are housed within the blender file. For rendering an object in the final image, it is positioned in predefined locations on the scene. Specifically, we designated ten empty positions, such as varied table locations in the scene. To make this object visible in the final image, the script will copy this object and place it in one of the ten positions.

Each question and answer group has a unique list of corresponding visuals used for image creation. The list of visible objects, which combines the correct-answer objects with an arbitrary 'noise' object selected outside the answer range, forms the objects that will be rendered in the scene. These visible objects will be rendered and displayed in the final image and they are tailored to match specific question-answer combinations, enhancing the necessity of image context for accurate question interpretation.

Blender's precision in object positioning is instrumental for visual reasoning. To craft images reflecting the visual reasoning in our questions, we start by defining a set of visual relations, e.g., right, left, front, between. These are integrated into our question matrix, enhancing questions with spatial references. For instance, the base query,"What are the foods that is unsuitable for vegans and free of fat?", evolves into "What are the foods to the left of the meat unsuitable for vegans and free of fat?".

Given the spatial relationship of "to the left of" and the anchor object "meat", we earmark one position to the anchor object (meat)'s right and nine to its left. We position one correct answer object to meat's right and fill the remaining left-side positions with other answers and 'noisy' objects. This approach challenges models to discern spatial relationships in their responses beside the logical reasoning we focused on.

## D   Answers Statistics

Figure 7 illustrates the answer length distribution based on the word count of answers in the LoRA dataset, which exhibits a wide and diverse range of answer lengths. Figure 7(a) shows the distribution of answer types, primarily encompassing five categories: individual object, multiple objects, number, yes/no, and none. The "none" category is designed to test logical reasoning skills, implying that the ground truth answer should be "none". For instance, after logical reasoning, it could be concluded that "none of the food in the image" satisfies the question's logical queries.

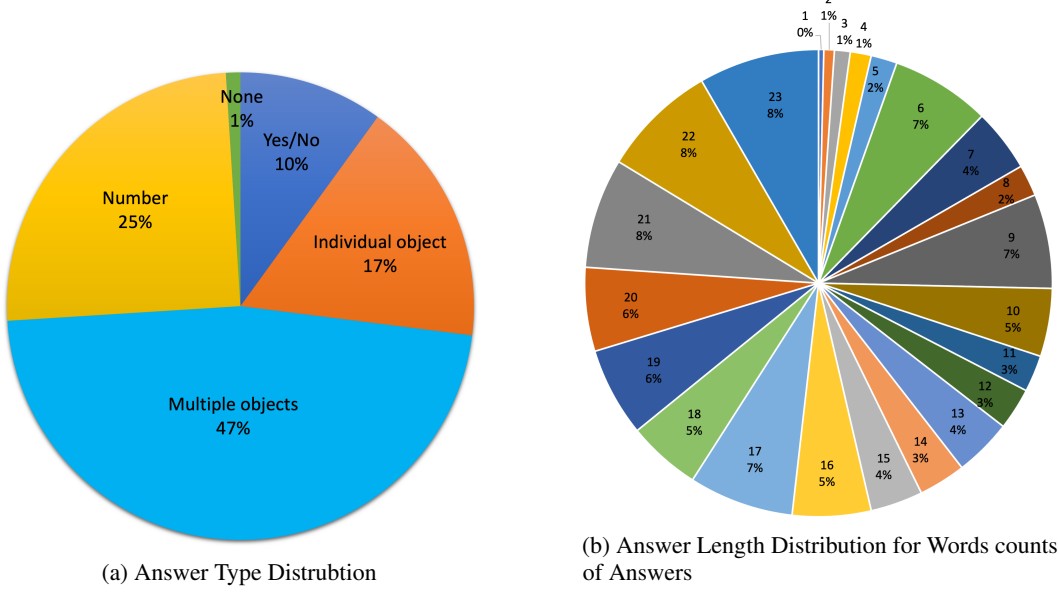

(a) Answer Type Distrubtion

(b) Answer Length Distribution for Words counts of Answers

Figure 7: Answer statistics.

# E Experiments

## E.1 Experiment Details

In Section 5 of the main paper, we conduct experiments involving various state-of-the-art VQA baselines along with large vision-and-language models.

**Input Size.** For VQA baselines, we use the maximum count of input words or tokens at 150.

**Batch Size.** The baselines of VQA are executed with batch sizes of 32.

**Zero-Shot and Few-Shot Baselines.** We iterate over our test dataset to evaluate cutting-edge large vision-and-language models. Specifically, we assess MiniGPT4, Multimodal-GPT, InstructBLIP, and Multimodal Chain-of-Thought (MMCoT) using our LoRA test dataset. For Multimodal Chain-of-Thought, we input the logical operators that are used to construct the questions as the prompt context.

The baseline evaluations are available in Baseline GitHub.

## E.2 Human Performance Study

The human performance study engaged a group of one hundred individuals, coming from varied careers, including students as well as professionals. They were tasked with answering logical challenges within the LoRA framework that span levels 1 through 3, requiring reasoning that ranged from a minimum of three steps to over nine.

In our evaluation methodology for each logical difficulty level, we provided potential answers including two formats: (i) open-ended and (ii) multiple-choice. In open-ended and multiple-choice formats, we expect exactly that. We measure the final accuracy of human answers based on the average score of human answers provided to the same level of logical questions, using the given metrics:

$$\text{accuracy} = \frac{\sum \text{scores of human answers}}{\# \text{ humans that provided that answer}} \tag{1}$$

In addition, we sought to assess the suitability of this logical complexity for human understanding. We curated a set of 50 sample questions and engaged humans reviewers to validate the logical complexity assigned to these questions. This process allowed for a rigorous and human-centered evaluation of the

logical complexity in our dataset. We discovered that humans struggle more with logical questions that involve more than two logical operators and logical expressions with negation logic. However, there is variability in logical reasoning skills among individuals. The sample questions, along with human reviews, are available in LoRA GitHub.

# F    LoRA Dataset Case Study

Figures 8 through 11 below demonstrate the LoRA dataset's zero-shot testing performance on MiniGPT4. It is evident that the latest large vision and language models are prone to logical errors. For instance, these models struggle to correctly interpret and answer questions involving negation. They often interpret a negation query as a standard query. For example, when asked, "not the same color as the wine in the glass", the model can correctly identify both the wine as red and the tomato as red, but struggles to reason that the question is asking for something that is not the same color. Furthermore, large vision and language models have a tendency to concoct answers that are either entirely unrelated to the questions or to generate generalized answers devoid of logical reasoning. This issue is not limited to the MiniGPT4 model; it also extends to other large vision and language models. Our experiments demonstrate that the LoRA dataset presents a significant challenge to models attempting to answer complex logical questions.

# G    Societal impact

The LoRA dataset is a self-constructed dataset based on the food and kitchen domain, designed to provide complex logical questions. The dataset does not include any user usage data or personally sensitive information. We have not identified any negative societal impact associated with the dataset.

# H    Limitations of our work

## H.1    Domain Specificity and Expandability

The food and kitchen domains serve as representative scenarios to demonstrate the intricacies of logical reasoning in VQA. While our focus is on logic problems related to the food-and-kitchen domain, the framework can be adapted to other domain ontologies.

Our approach and pipeline to create the dataset is generalizable. The framework utilizes the industry-standard OWL format for ontologies which is adaptable. It can work with any ontology (knowledge base) that adheres to the OWL specifications (standard W3C Web Ontology Language).

In addition, logical constructs employed by humans, such as conjunction, disjunction, negation, and conditionals, are consistent across diverse knowledge domains, such as food-kitchen, outdoor scenarios, or mathematics. The essence of logical inference remains unchanged irrespective of the content domain. Our dataset seeks to probe the performance of current VQA methods across varying logical complexities, rather than their capacity to leverage external knowledge.

## H.2    Logical Framework

Our logical reasoning model is rooted in a three-tiered structure, consistent with language sequence processing. This design stems from the observation that human language typically uses up to three logical operators.

## H.3    Image Generation Limitation and Expandability

For LoRA's image generation, we used Blender, an open-source software, due to its rich repository of realistic food-and-kitchen objects. It offers precise object positioning vital for visual reasoning. Emerging AI tools like stable diffusion promise broader image diversity, but they have current limitations in generating specific structured visuals, such as positioning an apple to the left of a banana and behind an orange. Our future work intends to incorporate these advanced generative tools to enhance our dataset further.

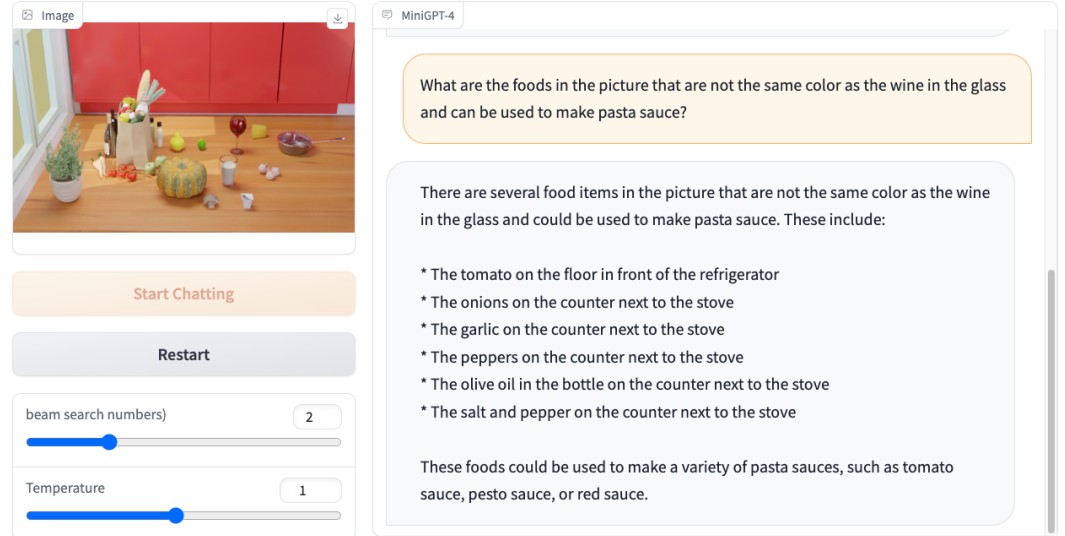

Figure 8: LoRA dataset's zero-shot testing performance on MiniGPT4. The correct answer should include a variety of foods such as garlic, mushroom, and cheese, but it should not include tomato. The case study demonstrates that the model lacks an understanding of the logic of negation and fails to infer the reasoning behind it. When asked what the colors of tomatoes and wine are respectively, the model gives both red. The model's answer merely describes an object in the image, but incorrectly provides the response "tomato sauce."

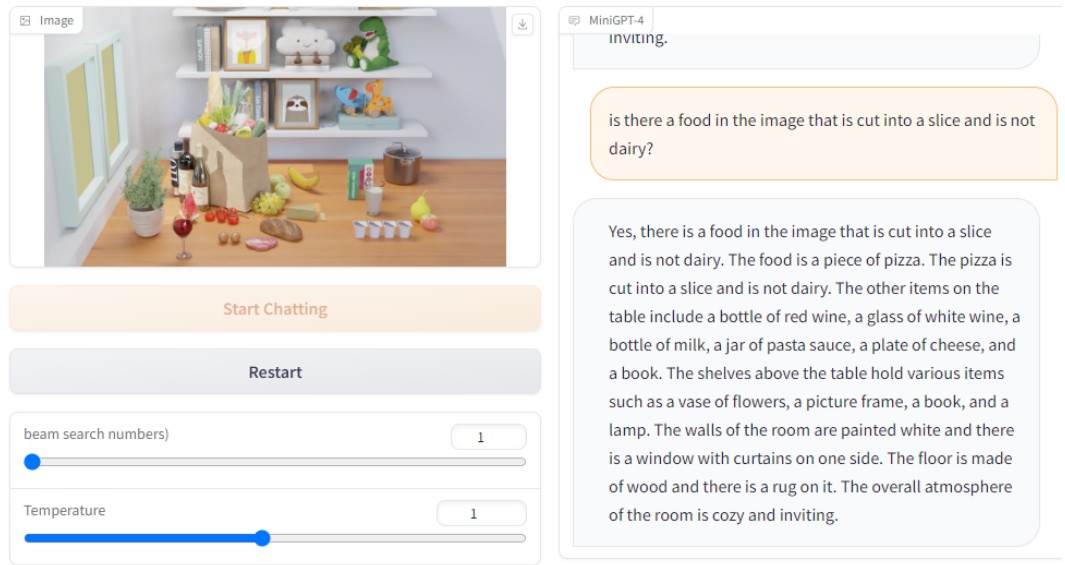

Figure 9: LoRA dataset's zero-shot testing performance on MiniGPT4. The question involves logical types such as **and**, **negation**, and **inclusion**, for example, the phrase "not dairy". The correct answer should be "yes, watermelon." However, the model incorrectly answers it as "pizza", which does not exist in the image. Furthermore, the model's response provides a description of the image rather than providing a reasoned answer.

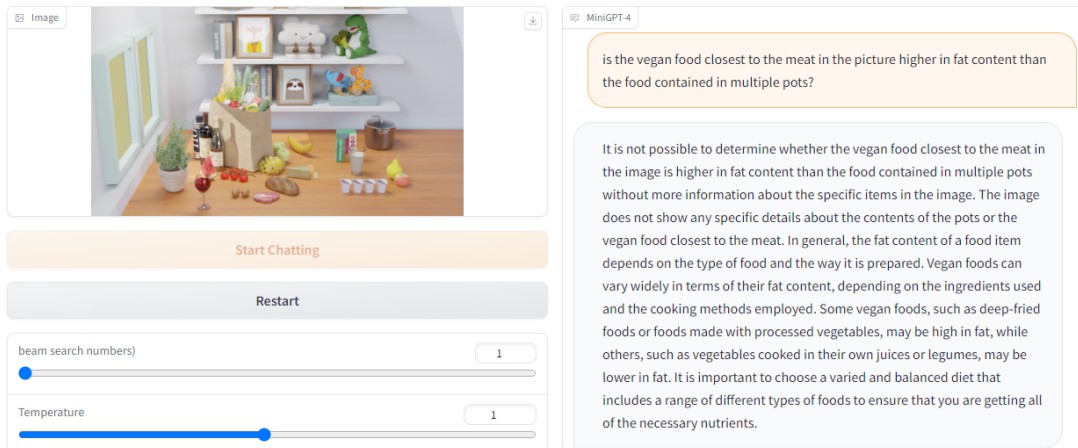

Figure 10: LoRA dataset's zero-shot testing performance on MiniGPT4. The question requires inclusive logical reasoning, first identifying if it is asking about vegan food, rather than other types of food. Then, visual information is needed to identify the visual relationship, specifically the object closest to meat, and to identify food contained in multiple pots. The correct answer is "no". However, the model failed to use commonsense knowledge to provide the reasoning answer.

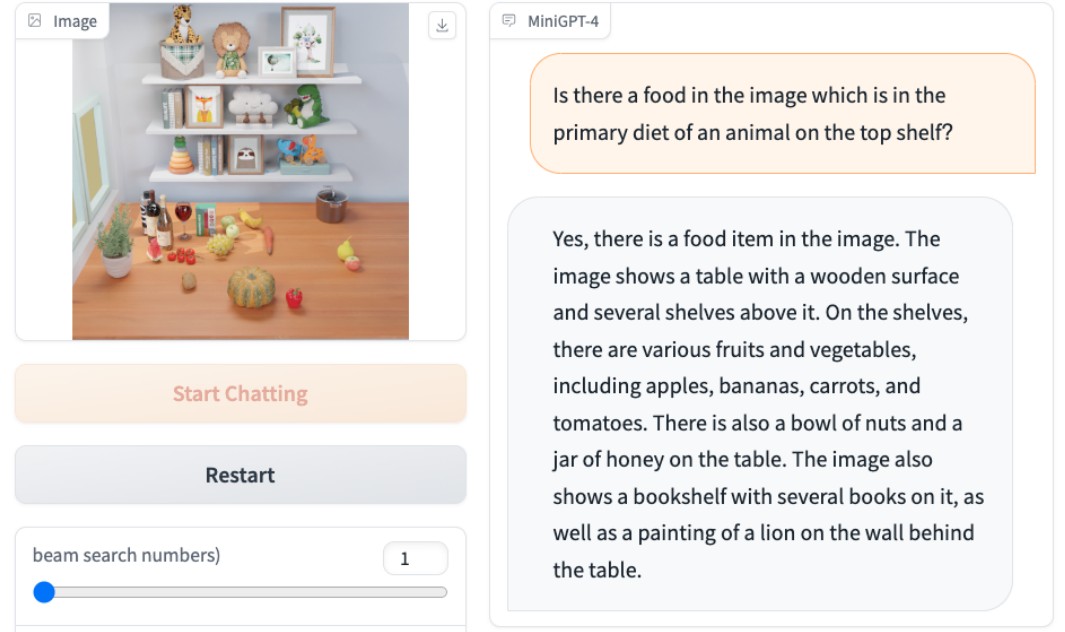

Figure 11: LoRA dataset's zero-shot testing performance on MiniGPT4. This is a verification question that requires both visual information to identify all the objects and commonsense knowledge to reason the answer that the animal's primary diet is meat, while the objects on the table are vegetables and fruit. The correct answer is "no". However, the model incorrectly answered "yes" and listed some objects in the image, making mistakes in the logical reasoning and visual relationships.