# OpenReview forum: "LoRA: A Logical Reasoning Augmented Dataset for Visual Question Answering"
_NeurIPS.cc/2023/Track/Datasets_and_Benchmarks — NeurIPS 2023 Datasets and Benchmarks Poster_

### Official Review · Reviewer_pVrF · 2023-07-04

**Rating:** 6
**Confidence:** 4

**Strengths:**

This work provides a valuable assessment of leading VQA and large vision-and-language models using the novel LoRA dataset. The dataset generation method, using Description Logic, ensures question diversity and complexity. The potential for researchers to customize the dataset using provided scripts is a substantial advantage, and the dataset's potential as a future benchmark is clear.

**Additional Feedback:**

This paper holds notable potential and with certain refinements, it could substantially contribute to the field of Visual Question Answering. I would encourage the authors to consider broadening the domain focus beyond food and kitchen to enhance the dataset's generalizability. An in-depth elaboration on the dataset generation process, specifically the ontology creation and question generation, would be beneficial. Reassessing the methodology used for determining question difficulty with a more rigorous approach, possibly with the inclusion of human validation, would increase the validity of the findings. Lastly, exploring alternative methods to the current reliance on third-party tools for key experimental stages could improve the work's independence, adaptability, and potential replicability. As a reviewer, I remain open to revising my score should the authors address the aforementioned points. This feedback is meant to be constructive, and I look forward to seeing the progress in the future revisions of this work.

**Clarity:**

The paper is generally well-written, although further detail on dataset generation methods and the complexity evaluation methodology would enhance the paper's clarity.

**Correctness:**

The claims made in the paper appear to be valid. However, the reliance on third-party tools and the narrow domain focus raise questions about the wider applicability and reproducibility of the results.

**Documentation:**

Although the provided scripts for customizing the dataset are a strength, the overall dataset documentation could be improved, especially regarding the ontology creation and question generation processes.

**Ethics:**

There do not appear to be any significant ethical concerns with this study.

**Limitations:**

The study's narrow domain focus may limit the generalizability of its findings. Insufficient detail on dataset generation and the use of third-party tools for significant experimental stages could potentially limit the work's scalability and reproducibility. The methodology for determining question complexity could be more comprehensive and rigorous.

**Opportunities For Improvement:**

The work's primary reliance on a specific domain (food and kitchen) raises concerns about the generalizability of results across diverse fields. Greater elaboration on the dataset generation methods and ontology creation process would improve clarity. The methodology for gauging question difficulty requires further justification and could benefit from a more rigorous approach. The heavy dependence on third-party tools for key experimental stages may limit the study's scalability and adaptability.

**Relation To Prior Work:**

The authors have effectively differentiated their work from previous contributions, primarily through the introduction of the novel LoRA dataset and its application to VQA.

**Summary And Contributions:**

The authors present a novel contribution to the field of Visual Question Answering (VQA) through the LoRA dataset, designed to foster complex logical reasoning. Their unique method for automated generation of complex questions and images sets this work apart. While the study provides significant insights into the capabilities of current VQA and vision-and-language transformer models, there are areas for improvement and expansion.

---

> ### Author Response · Authors · 2023-08-20
> **Responses to Reviewer pVrF**
>
> Dear reviewer, thank you very much for your valuable feedback. We appreciate your recognition of our paper's novelty, which emphasises diverse, complex, and layered logical reasoning. We are also encouraged that our provided automated scripts offer the potential to substantially benefit researchers by customizing the dataset. We appreciate the positive outlook on our work, regarding our dataset's potential as a future benchmark.
>
> We address the questions below:
>
> **Q1: Domain Specificity and Expandability**
>
> We acknowledge the concern raised regarding the focus on the kitchen and food domains.
>
> The food and kitchen domains serve as representative scenarios to demonstrate the intricacies of logical reasoning in VQA, not to limit our study's scope.
>
> Our approach and pipeline to create the dataset is generalizable. This is because the framework we utilized for ontology is based on the industry-standard OWL format which is adaptable. It can work with any ontology (knowledge base) that adheres to the OWL specifications, which is the standard W3C Web Ontology Language. (As detailed in Section 3.3.1, lines 165-168: "Our algorithm can unroll any ontology that is based on or can be converted to owlready2, not only our customised ontology". ).
>
> Moreover, logical constructs such as conjunction, disjunction, negation, and conditionals are universal across various knowledge domains, such as food-kitchen, outdoor scenarios, or mathematics. The essence of logical inference remains unchanged irrespective of the content domain. Our dataset seeks to probe the performance of current VQA methods across varying logical complexities, rather than their capacity to leverage external knowledge.
>
> **Q2. Reliance on Third-Party Tools:**
>
> **a. Blender for Image Generation:**
>
> The third-party tool to generate realistic images for the LoRA dataset is based on Blender, which is an open-source free software platform, offering transparency and reproducibility. It provides a rich repository of realistic food-and-kitchen objects and it also provides objects from other domains, allowing for expansion.
>
> Blender offers precise object positioning, which is vital for visual reasoning. Emerging AI tools like stable diffusion promise broader image diversity, but they currently fall short in creating specific structured visuals, such as positioning an apple to the left of a banana and behind an orange. Our future work intends to incorporate these advanced generative tools to enhance our dataset further.
>
> Blender allows automated script writing (Python) to place objects in the scene automatically and generate realistic images. All assets used in our study, including Python scripts and Blender backgrounds to food-and-kitchen objects, are stored in a single Blender file.
> We have publicly released the Blender file in our LoRA GitHub repository. To initiate a new image generation, simply execute the provided Python scripts.
>
> **b. OWL and Ontology Tools:**
>
> Our decision to align with the OWL format ensures wide compatibility and adaptability. Most modern ontology tools and libraries support OWL, making our approach easily reproducible and scalable. The ontology we used and the scripts how to unroll ontology and details have been released on LoRA Github. We use OWLReady2 which is a python script to write Ontology.
>
> **Q3: Further detail on dataset generation methods and the complexity evaluation methodology**
>
> We appreciate this valuable suggestion. We have added further details to the revised Supplementary material. In Section C titled 'Dataset Generation Additional Details', we have included subsections C.1 'Ontology Creation Additional Details' (Line 117-136), C.2 'Question Generation Additional Details' (Line 137-170), and C.3 'Image Generation Additional Details' (171-200) to enhance clarity.
>
>
> **Q4: Human validation of the question difficulty.**
>
> We appreciate the insightful suggestion from the reviewer. Firstly, we generated questions using templates, ensuring control over the known logical complexity based on formal Description Logics. Additionally, our original work includes a human performance study on different levels of question logical difficulties, detailed in Section E.2 'Human Performance Study' in the supplementary materials. Furthermore, we have also taken the following action: We curated a set of 50 sample questions and engaged human reviewers to validate the logical complexity assigned to these questions. We sought to assess the suitability of this complexity for human understanding. We discovered that humans struggle more with logical questions that involve more than two logical operators and logical expressions with negation logic. However, logical reasoning skills vary among individuals. The sample questions, along with the reviews, are available in our GitHub repository. This process allowed for a rigorous and human-centered evaluation of the logical complexity in our dataset.

---

> > ### Comment · Reviewer_pVrF · 2023-08-28
> >
> > Thank you to the authors for their rebuttal and efforts to address the concerns raised. After careful consideration of the provided clarifications and improvements, I have increased my rating.

---

> > > ### Author Response · Authors · 2023-08-29
> > > **Thank you for reviewer's comment**
> > >
> > > Thank you very much for your valuable feedback, which has helped us improve our paper. We appreciate the opportunity to address the concerns. Thank you so much for increasing your rating and we are encouraged by the recognition of our efforts.

---

### Official Review · Reviewer_KjfL · 2023-07-21
**Submission 480 review**

**Rating:** 7
**Confidence:** 3
**Clarity:** The paper is well-written and well-or…

**Strengths:**

The paper analyzes the logical reasoning in previous datasets and defines complexity as the number of logic types besides the number of operators.

Based on the proposed definition, the authors propose a reasonable way to construct the questions with much more types of logic components than previous methods.

The paper provides the generation scripts that can contribute to the community for generating different samples.

**Additional Feedback:**

Line 174: ⟨ sentence block 3 ⟩ -> ⟨ sentence block 1⟩

Also see weakness.

**Correctness:**

The dataset construction process is sound. The evaluation is performed correctly.

**Documentation:**

The paper gives sufficient detail on data collection, URL, and corresponding license.

**Ethics:**

The dataset is automatically generated and does not have other ethical concerns.

**Limitations:**

The authors adequately addressed the limitations and potential negative societal impact.

**Opportunities For Improvement:**

The authors first generate the question, then the answer and image. The paper does not give the details of how to make the correct answer unique if and only if given the corresponding image, such that the model can not answer the question with text only.

**Relation To Prior Work:**

The paper clearly discussed the difference from previous works.

**Summary And Contributions:**

The paper proposed a new VQA dataset whose question requires much more complex logical reasoning than previous works. It first shows that the previous VQA datasets only have conjunction, disjunction, and negation, then it constructs the questions to require the compound of conjunction, disjunction, and negation; exist and universal restriction; concept and role hierarchy, etc. To generate the dataset, the authors define a kitchen domain-specific ontology and logic operators, then randomly select 1~3 <roles, entities> and use logic operators to compose them as a question. Lastly, authors generate diverse textual questions with templates, retrieve the answer, and render the final image.

---

> ### Author Response · Authors · 2023-08-20
> **Responses to Reviewer KjfL**
>
> Dear reviewer, thank you very much for your valuable feedback and we appreciate your insightful observation. We address your concerns below.
>
> **Q1: Unique answers given the corresponding image.**
>
> Our methodology guarantees the uniqueness of each question-answer-image triplet. After generating the questions and their correct answers, we create images based on these correct answers, with the inclusion of random 'noisy' objects which are not in the correct answer range. This ensures that the image context is crucial for correctly interpreting the question and identifying the accurate answer. The model cannot derive the answer from the question text alone due to the added complexity of the 'noisy' objects in the image.
>
> The details of this procedure can be found in Section C.3, Image Generation Additional Details (Line 176 - 189), of the revised supplementary material.
>
> **Additional Feedback**
>
> Line 174: ⟨ sentence block 3 ⟩ -> ⟨ sentence block 1⟩ , thank you very much for this insightful observation. We have revised this typo error and updated it in the revised paper.

---

> > ### Comment · Reviewer_KjfL · 2023-08-30
> >
> > Thanks for the response. I have no further concerns and will keep the rating unchanged.

---

> > > ### Author Response · Authors · 2023-08-30
> > > **Thank you for reviewer's comment**
> > >
> > > Thank you very much for your feedback and valuable time throughout the review process.

---

### Official Review · Reviewer_3Fyb · 2023-07-21
**A dataset for logical reasoning in VQA**

**Rating:** 6
**Confidence:** 2

**Strengths:**

Large-scale dataset. The paper introduces LoRA, a fresh VQA dataset comprising 200,000 intricate and diverse logical reasoning questions, based on the formal Description Logic SROIQ. It overcomes existing dataset limitations and facilitates a comprehensive evaluation of VQA models' logical reasoning abilities.

Evaluation metrics. With progressively challenging questions requiring multi-step logical reasoning and various logical inferences, LoRA presents a unique evaluation setup that tests the state-of-the-art VQA models' performance in handling complex inference and reasoning tasks.

Contribution to research community. The paper provides automated scripts to generate logical questions, encouraging the expansion of logical reasoning datasets. Additionally, it introduces a formal definition of logical difficulty based on Description Logic, systematically evaluating VQA and large vision-and-language models, and highlighting promising avenues for future research.

**Additional Feedback:**

N/A

**Clarity:**

The paper exhibits great clarity in its presentation. It highlights the significance of LoRA in addressing the limitations of existing VQA datasets and providing a comprehensive evaluation of logical reasoning capabilities. Additionally, incorporating diverse logical complexities and offering comprehensive guidelines for generating custom datasets would strengthen its impact on the research community. Overall, the paper's clear and concise writing style contributes to its accessibility and understanding of the proposed logical reasoning augmented VQA dataset.

**Correctness:**

The paper is sound and correct. The evaluation metrics are fair and appropriate.

**Documentation:**

The documentation is clear, complete and easy to follow.

**Limitations:**

The proposed LoRA dataset, although comprehensive and challenging, focuses solely on formal and complex logical reasoning within the context of food-and-kitchen scenarios. As a result, the evaluation may not fully capture the broader capabilities of VQA models in handling logical reasoning across various domains and real-world situations.

While the paper highlights the performance of state-of-the-art VQA models on the LoRA dataset, it does not extensively explore potential solutions or strategies to improve their logical reasoning capabilities. Further investigations into model architectures, training techniques, or multimodal fusion methods specifically tailored for complex logical inference could provide valuable insights for advancing VQA models in this aspect.

The paper mentions the logical difficulty levels of the questions based on Description Logic, but it may not fully encompass all possible facets of logical complexity. Other formal logical frameworks or real-world complexities may exist that are not fully accounted for in the current evaluation.

**Opportunities For Improvement:**

**Real-world Diversity.** While LoRA presents a significant improvement in logical reasoning evaluation, it focuses on food-and-kitchen scenarios. Expanding the dataset to encompass a broader range of real-world scenarios, such as outdoor scenes, industrial settings, or social interactions, would enhance its applicability and generalize the evaluation to a wider context.

**Incorporating Uncertainty.** The current dataset assumes a deterministic world, which may not accurately represent real-life ambiguity and uncertainty. Including questions that involve uncertain information, probabilistic reasoning, or unknown variables would make the evaluation more realistic and challenge models to handle uncertain situations effectively.

**Human Performance Benchmarking.** The paper primarily compares VQA models' performance on LoRA, but it lacks a comprehensive benchmark of human performance on the dataset. Integrating human performance metrics would provide valuable insights into the gap between human and machine reasoning capabilities and offer a stronger basis for measuring model advancements.

**Relation To Prior Work:**

Prior works are clearly discussed in this paper.

**Summary And Contributions:**

The paper presents LoRA, a novel Logical Reasoning Augmented VQA dataset designed to evaluate formal and complex logical reasoning capabilities of VQA models. It contains 200,000 diverse questions generated using strong programs based on SROIQ Description Logic, accompanied by realistic kitchen scenes and ground truth answers. LoRA sets a new benchmark for logical reasoning evaluation, enhancing the development of VQA models in tackling challenging reasoning problems.

---

> ### Author Response · Authors · 2023-08-20
> **Responses to Reviewer 3Fyb**
>
> Dear reviewer, thank you for your valuable feedback. We appreciate the positive recognition of our work, including our dataset's expansive scale, comprehensive and unique evaluation metrics to evaluate complex and layered logical reasoning in VQA. We are encouraged by the recognition of our automated scripts, which can contribute to expanding customized datasets for researchers.
>
> We address the questions below:
>
> **Q1: Real-world Diversity.**
>
> The food and kitchen domains serve as representative scenarios to demonstrate the intricacies of logical reasoning in VQA. The logical constructs that humans employ, like conjunction, disjunction, negation, and conditionals, remain consistent across different domains of knowledge, such as food-kitchen, outdoor scenarios, or mathematics. The core part of logical inference would be the same regardless of the content. This is what our dataset would like to explore: how well the current VQA methods can perform on different levels of logical complexity, rather than their capacity to leverage external knowledge.
>
> Secondly, our approach and pipeline to create the dataset is generalizable and extendable. This is because the framework we utilized is the industry-standard OWL format for ontologies which is adaptable. It can work with any ontology (knowledge base) that adheres to the OWL specifications, which is the standard W3C Web Ontology Language. (refer to Section 3.3.1, lines 165-168: "Our algorithm can unroll any ontology that is based on or can be converted to owlready2, not only our customised ontology". ).
>
> **Q2: Incorporating Uncertainty.**
>
> We agree that incorporating uncertainty into our dataset would make the evaluation more realistic and challenging. This is indeed one of our future directions, as we plan to extend our dataset with more questions that involve uncertain information, probabilistic reasoning, or unknown variables. We believe that this would further test the logical reasoning capabilities of VQA models and push them to handle real-world ambiguity and uncertainty effectively.
>
> We agree that Description Logic may not be able to capture some types of uncertainty, ambiguity, or inconsistency that may occur in other logical frameworks or real-world complexities. However, the current VQA methods and the latest multi-modal large language models still struggle to handle the levels of logical complexity that our dataset presents.
>
> **Q3: Human Performance Benchmarking.**
>
> Human performance benchmarking was provided in the original supplementary document, under Section E.2 Human Performance Study. Table 2 of the supplementary document presents the human performance benchmarking results. Additionally, the human performance results have been incorporated into Table 5 of the main paper.
>
> **Q4: Exploration of potential solutions or strategies to improve their logical reasoning capabilities.**
>
> Our primary focus in this paper is introducing a new dataset. Strategies to enhance logical reasoning in VQA are discussed in our separate work, "A Symbolic Neural Reasoning Model for VQA". We kindly direct reviewers to that paper for insights on potential solutions.

---

### Official Review · Reviewer_JcBn · 2023-07-21
**Interesting dataset for more complex logic-based VQA**

**Rating:** 6
**Confidence:** 3

**Strengths:**

- The work seeks to explore an important ability of complex logic-based visual question answering, which previous works/datasets have not adequately studied.

- The dataset creation mechanism is well described and the consideration of different logic operations (based on description logic) is adequate, covering a more extensive set of primitive operations (e.g. combination of multiple conjunctions/disjunctions/negations in addition to restrictions, transitivity, etc) than previous works have explored.

-  The dataset is also stated to provide logical programs and associated steps to better analyze models and assess their reasoning abilities. It also also analyzing models in terms of question syntax complexity and answer inference complexity, and hence provides a more comprehensive evaluation of models for diverse question/reasoning types as well as difficulty levels.

- The considered models for evaluation are adequate/relatively recent and results are reported for multiple question complexities to highlight possible limitations of existing models.

**Additional Feedback:**

NA

**Clarity:**

Yes. Minor typos: L2 ('locigal'), table 7 (4-5 complex inference steps: not sure if the symbol b/w C and D is correct?)

**Correctness:**

The dataset construction and evaluation of baseline largely seems correct. There is concern whether the dataset can be solved by an external knowledge-integrated model and currently does not adequately test visual-logical reasoning.

**Documentation:**

Data collection is described adequately and the dataset is publicly available. Authors could consider releasing baseline evaluation code for reproducibility.

**Limitations:**

- The authors mention in supplemental checklist (L196) that they did error analysis in sec.5. However, this is the baselines evaluation and not limitations of the authors' work. Hence, limitations do not seem to be adequately noted/addressed.

**Opportunities For Improvement:**

1. A major limitation is that the questions appear to require more language/knowledge-based logical reasoning rather than visual reasoning. For example, taking some of the provided example questions - "If we do not have milk, is there another dairy product that does not necessarily contain fat but is rich in protein that can be substituted for breakfast?" -- here this seems to require a language or knowledge-based model to first infer possible candidates (without looking at the visual scene), which once computed, can then be simply solved through object detection (and not exactly 'visual reasoning'). In contrast, consider a question based on CLEVR -- "How many small spheres are there to the left of the cylinder and right of the shiny cube?", here the computation seems to require both logical and visual reasoning. Hence, authors could better highlight whether and how the proposed dataset particularly requires visual-logic reasoning to make a stronger claim as a complex logic VQA dataset.

2. In the reported baselines, it could be beneficial to report the performance of external knowledge-integrated models (perhaps by utilizing foodKG). Currently as per L285, it seems no external knowledge base is provided to the baseline models. Having these results could better indicate how much of the task is solvable through external knowledge and how much requires visual reasoning.

3. For the MAC architecture, it is unclear whether when authors say 'finetuned', is the CLEVR-trained model finetuned further?  More generally, details of how baselines are applied for the dataset could be described more clearly or released publicly for reproducibility.

**Relation To Prior Work:**

Yes, largely so. Authors could consider differentiating from relevant recent datasets such as "QLEVR: A Diagnostic Dataset for Quantificational Language and Elementary Visual Reasoning" (NAACL 2022).

**Summary And Contributions:**

The paper proposes a novel dataset termed LoRA (Logical Reasoning Augmented Dataset) to improve and evaluate specifically logic-based reasoning of visual question answering (VQA) models. In contrast to existing VQA datasets, LoRA utilizes a specialized description logic to formulate image-question-answer pairs of higher logic reasoning complexity with various primitive logic operations composed at increasing levels. The dataset specifically utilizes the FoodKG knowledge graph and is currently limited to kitchen/food-based scenes (but can be extended to other domains). The authors propose a multi-step process that utilizes an ontology, description logic formalisms and a rule-base for automatic question and answer generation besides using Blender to render images. The dataset contains 200,000 questions for over 100,000 kitchen scenes and is decently sized for training, validation and testing.

---

> ### Author Response · Authors · 2023-08-20
> **Responses to Reviewer JcBn**
>
> Dear reviewer, thank you for your valuable feedback. We appreciate your recognition that our dataset: 1) focuses on complex logic-based VQA and layered logical complexities, which previous datasets have not adequately studied. 2) the automatic dataset creation mechanism is adequate and well-described. 3) provides a comprehensive and recent evaluation for diverse question/reasoning types as well as difficulty levels.
>
> We address the questions below:
>
> **Q1: Concerns regarding Visual Reasoning in LoRA**
>
> The questions within the LoRA dataset extend beyond mere knowledge-based reasoning or simple object detection. It's imperative to emphasise that they necessitate an integration of four data sources: visual, linguistic, external knowledge and logical reasoning.
>
> The visual aspect is not limited to straightforward object detection in our dataset. It encompasses layers of visual reasoning, as illustrated by the example in LoRA, ''Can we use the food **between** eggs and bread to make a meal for vegetarians?'' referenced in Figure 1 on page 2. Consider another example in LoRA: ''Is the item **to the left of** the meat unsuitable for vegans and free of fat?'' in supplementary Table 1. These questions in LoRA combine visual reasoning, commonsense knowledge, and logical reasoning.
>
> In the supplementary Table 1 presents an extensive array of questions, the data these questions rely upon are represented by the 'Feature' column (V,Q,KB), where 'V' indicates the necessity for visual information in tandem with the question and knowledge base. In the revised version of the supplementary material, Section C.3 (Lines 190-200) is added to detail the process of crafting images that reflect the visual reasoning in our questions.
>
> **Q2: Suggestion to Include External Knowledge-integrated Baseline Model**
>
> We opted for multi-modal large language models over traditional knowledge-integrated VQA models, primarily due to their advanced capabilities because they leverage large-scale external knowledge data. These LLMs are trained on extensive data, encompassing comprehensive food domain knowledge and other domain knowledge. Therefore, we can more clearly evaluate if they can handle higher-level logical reasoning problems based on they already have enough external domain knowledge.
>
> The baseline evaluation is to probe the performance of current VQA methods across varying logical complexities, rather than their capacity to leverage external knowledge or visual reasoning.
>
> **Q3: Clarification on MAC Architecture Fine-Tuning and Baseline Evaluation Reproducibility.**
>
> Thank you for this suggestion. To clarify, we did not fine-tune the MAC architecture. We implemented the MAC architecture on our custom dataset, LoRA, without further fine-tuning from its CLEVR-trained state. It is an end-to-end training. We have revised the paper in 298 -302 for Mac baseline.
>
> The baseline evaluation code has been released to GitHub detailed in Supplementary Section E Line 227.
>
> **Q4: Limitations of our work:**
>
> Thank you for this constructive feedback. We have added an additional limitation of our work in supplementary section H. Limitations of our work (Line 260 -283).
>
> **Q5: Minor typos: L2 ('locigal') and conjunction symbol: C ⊓ D**
>
> We have revised minor typo of locigal.
>
> The conjunction and disjunction symbols between C and D is correct symbol based on Description logic (reference citation: [2])
>
> **Q6: Comparison with QLEVR dataset**
>
> Logical reasoning differs from visual reasoning, which often revolves around counting or interpreting spatial relationships in visual scenes. For example, questions from QLEVR or CLEVR, like ''Are all the cyan metallic triangular prisms on the brown plane?'' or ''How many small spheres are to the left of the cylinder and right of the shiny cube?'', mainly rely on spatial understanding and basic logical concepts like universal quantification or conjunction. However, they lack advanced constructs like conditional reasoning.
>
> In contrast, our LoRA dataset delves into intricate logical constructs. Consider the LoRA question: "If we don't have milk, is there another dairy product that is potentially fat-free but protein-rich for breakfast?" This isn't just a simple query or visual pattern recognition. It requires understanding conditionals, negations, existential constraints, and concept hierarchies, demonstrating the depth of logical reasoning needed.
>
> **Richness of Visual Content:** Moreover, while QLEVR and CLEVR are both restricted to geometric imagery, LoRA uses diverse, realistic kitchen and food visuals, closely mirroring real-world scenarios.

---

> > ### Comment · Reviewer_JcBn · 2023-08-29
> >
> > Thank you for the detailed response and clarifications.
> >
> > However, my primary concern still remains that the current dataset is less complex from a visual reasoning perspective and more complex from a knowledge-based reasoning perspective. In the given examples, ''Can we use the food between eggs and bread to make a meal for vegetarians?'' and ''Is the item to the left of the meat unsuitable for vegans and free of fat?'', the visual reasoning steps seem to be between 1-3 steps (which is relatively low from existing datasets such as CLEVR [1] and PTR [2]). For the first question, it seems a model needs to identify bread and eggs, and then the food between them. After which, it is knowledge-based reasoning ('make meal for vegetarians'). Similarly, for the second question, the visual reasoning step is identifying objects 'to left of meat', after which it is knowledge-based reasoning (filtering candidate objects based on properties of 'unsuitable for vegans' and 'free of fat').
> >
> > Further, I understand that baseline models are chosen from existing VQA models; however, I still believe considering a simple knowledge-augmented model and reporting its performance could better highlight the contributions/difficulty of the proposed dataset. (E.g. if an external knowledge-augmented model is still not able to obtain high performance, it's a stronger indication that the visual reasoning involved is indeed complex and not solvable through mere external knowledge integration).
> >
> > I am borderline positive about this work as existing datasets (such as CLEVR, PTR, QLEVR) are limited to synthetic environments while real-world datasets such as GQA are relatively less logically complex. Hence, I have raised my score to 6. However, I encourage authors to include more detailed differentiation from these datasets. And further, to show performances of an external knowledge-augmented model to better highlight complexity of the dataset.
> >
> >
> > [1] https://openaccess.thecvf.com/content_cvpr_2017/papers/Johnson_CLEVR_A_Diagnostic_CVPR_2017_paper.pdf
> > [2] https://proceedings.neurips.cc/paper/2021/file/918f5cd5a5c0d48671d4d4fc54bab2e9-Paper.pdf

---

> > > ### Author Response · Authors · 2023-08-30
> > > **Thank you for reviewer's comment**
> > >
> > > We thank the reviewer for the insightful comments. We greatly appreciate the additional feedback and the increase of score, as well as the recognition that our work is filling a gap in real-world logical reasoning datasets with extra complexity.
> > >
> > > We understand the concerns raised about visual versus knowledge-based reasoning. Previous work on VQA has predominantly concentrated on direct or explicit observations and visual relationships. These visual relationships, such as "bigger," "smaller," "larger," or "of the same size," "at the back of," "the same color" in CLEVR, PTR, represent visual characteristics rather than true logical reasoning. We aim to emphasize the intricacies of logical reasoning in VQA.
> > >
> > > In the question "Is the item to the left of the meat unsuitable for vegans and free of fat?", the visual reasoning involves identifying objects 'to the left of the meat', and subsequent knowledge-based reasoning filters the candidate objects based on 'unsuitable for vegans' and 'free of fat'. However, our core focus is on the intertwined logical reasoning. In this question, there are three logical constructs at play: NOT, AND, NOT. The challenge is not merely in leveraging knowledge, but in navigating these logical intricacies to derive an accurate answer.
> > >
> > > Regarding using an external knowledge-augmented model, we concur about its potential insights. Our intent with large language models was to gauge how well they perform when faced with logical complexities since they're inherently knowledge-rich. We would like to consider this suggestion for future evaluations.

---

### Official Review · Reviewer_G9pJ · 2023-07-22
**LoRA: A Logical Reasoning Augmented Dataset for Visual Question Answering**

**Rating:** 7
**Confidence:** 3
**Correctness:** Yes
**Clarity:** Yes

**Strengths:**

* The proposed dataset is flexible and expandable, yet still a challenging reasoning task for VQA models
* The proposed dataset has test cases with different levels of difficulty, which can be a good measure of the VQA model's capabilities
* Provides an automated approach from extracting the knowledge base to generating data images
* Experiments with a broad range of models and different scenario (finetuning, zero-shot, few-shot)

**Additional Feedback:**

N/A

**Documentation:**

see opportunities for improvement

**Opportunities For Improvement:**

* One of my concerns is about the automatic framework. In Figure.2 the authors mentioned that "Begin with a manually constructed knowledge base" and in section 3.1. the authors give a particular form that the knowledge base should follow. Does it mean that this automatic framework is not available for all knowledge bases?
* I have a slight concern that the experimental part just simply describes the data without analyzing it in depth. Also, given the presence of \[1\]\[2\]\[3\], some conclusions such as "the importance of integrating visual, question, and knowledge base information for effective VQA reasoning" and some pre-trained model can't solve difficult reasoning problem is not that novel. As a result, the "revealed important directions for future research" described in the contribution is not that clear.
* At the same time, for reproducibility reasons, I think that part of the experiment scenarios such as zero-shot needs to carefully describe the input template of the experiment, as the form of prompt has an impact on the effectiveness of the language model.

[1] Ding Y, Yu J, Liu B, et al. Mukea: Multimodal knowledge extraction and accumulation for knowledge-based visual question answering[C]//Proceedings of the IEEE/CVF Conference on Computer Vision and Pattern Recognition. 2022: 5089-5098.

[2] Schwenk D, Khandelwal A, Clark C, et al. A-okvqa: A benchmark for visual question answering using world knowledge[C]//European Conference on Computer Vision. Cham: Springer Nature Switzerland, 2022: 146-162.

[3] Valmeekam K, Olmo A, Sreedharan S, et al. Large Language Models Still Can't Plan (A Benchmark for LLMs on Planning and Reasoning about Change)[J]. arXiv preprint arXiv:2206.10498, 2022.

**Relation To Prior Work:**

see opportunities for improvement

**Summary And Contributions:**

Summary:

This paper proposes a new visual question answering dataset, LoRA, and a framework for automatically generating VQA datasets with different logical relations. In contrast to previous datasets, LoRA contains more complex and multi-step reasoning samples. Experiments reveal that several state-of-the-art VQA models are unable to solve difficult logical reasoning problems.

Contributions:

1. A novel, expandable and challenging VQA dataset
2. Systematically evaluates the logical reasoning capabilities of existing VQA models and large-scale visual language models at different levels of complexity, revealing directions for future research

---

> ### Author Response · Authors · 2023-08-20
> **Responses to Reviewer G9pJ**
>
> Dear reviewer, thank you for your valuable feedback. We appreciate your recognition of our dataset's 1) novelty, expandability, and challenge in VQA, 2) our provision of varying logical difficulty questions, 3) the automated approach for knowledge base extraction to image generation, and 4) our systematic evaluation of current VQA models. We address your questions below:
>
> **Q1: Is the automatic framework available for all knowledge bases?**
>
> Our dataset creation approach is generalizable and available for other knowledge bases. The framework that we utilized to construct the ontology is based on the industry-standard OWL format. It is compatible with any ontology (knowledge base) that aligns with OWL specifications - the standard W3C Web Ontology Language.
>
> When we state "Begin with a manually constructed knowledge base", and “particular form that the knowledge base should follow”, it refers to the initial setup where domain-specific knowledge needs to be structured into an ontology and the form is based on the standard OWL format. Our algorithm can 'unroll' any standard ontology, converting its structured data into table dataframes for flexible processing. As mentioned in Section 3.3.1 (line 165-168): “Our algorithm can unroll any ontology that is based on or can be converted to owlready2, not only our customised ontology.
>
>
> **Q2: The in-depth analysis for experiment:**
>
> Thank you for the valuable suggestion to enhance our paper. Additional in-depth analysis of models’ different behaviors and performance has been added to delve deeper into the underlying reasons, elucidating factors that contribute to the strengths or weaknesses of the models. This has been detailed in the added subsection 5.2 Error Analysis, lines 335 - 358 in the revised paper.
>
>
> **Q3: Comparisons to existing work [1][2][3] and Novelty of our work.**
>
> We appreciate the reviewer providing additional references.
>
> While [1] and [2] delve into knowledge-based VQA problems, and our dataset focuses on Logical Reasoning.
>
> Logical reasoning is different from purely knowledge-based reasoning, which requires external knowledge to provide a direct answer. Instead, logical reasoning involves navigating complex logical constructs.
>
> **Comparison Examples:**
> The examples provided from [1] and [2], such as "What type of architecture is shown in these buildings?" or "How many people will dine at this table?", primarily require external knowledge or visual recognition to give direct answers. They do not necessitate formal and complex logical reasoning.
>
> In contrast, LoRA focuses on intricate logical structures such as negation, conditionals (if...then), etc, 9 types of formal logic and combination of them. It encompasses a multi-step logical process that cannot be answered by merely querying an external knowledge base or recognizing visual patterns.
> An example question from LoRA, "**If** we **do not** have milk, is there another **dairy** product that **does not** necessarily contain fat **but is rich in** protein that **can be substituted** for breakfast?". This isn't just a simple query, it requires understanding conditionals, negations, existential constraints, and concept hierarchies, demonstrating the depth of logical reasoning needed.
>
> **Contribution Clarity**: As to the novelty of our conclusions, it is precisely this emphasis on structured, hierarchical logical reasoning that differentiates LoRA from other VQA datasets. Our aim is not merely to underscore the importance of integrating visual, question, and knowledge base information but to emphasize the necessity for models to reason through complex, layered logical constructs.
>
> **Granularity of Logical Complexity**: In addition, LoRA offers a systematic categorization of questions based on their logical complexity, ranging from simpler logical constructs to more convoluted ones (Please refer to Table 1 in the supplementary document.) This granularity is absent in the mentioned references.
>
> Regarding [3], it is imperative to highlight that the focus is on LLMs' capabilities in planning, rather than on logical reasoning in the context of visual questions. Specifically, while [3] investigates planning and reasoning about change in the domain of LLMs, it doesn't delve into visual question logical reasoning problems.
>
> **Q4: What is the zero-shot experiment input prompt?**
>
> For zero-shot experiments, we used the questions from the LoRA dataset as the prompts, with the accompanying images serving as the input images for these models. Each question in the dataset not only guided the model in the direction of the answer but also implicitly highlighted the logical construct being evaluated.
> This additional description has been added in the revised paper: Refer to paper Section 5.1 line 314-317.

---

> > ### Comment · Reviewer_G9pJ · 2023-08-31
> >
> > Thanks authors for the efforts for the author response. The newly added explanations have addressed my previous concerns. I believe these results will improve the quality of the submission. As such, I would raise my score.

---

> > > ### Author Response · Authors · 2023-08-31
> > > **Thank you for reviewer's comment**
> > >
> > > We sincerely appreciate your valuable suggestions and feedback, which have helped us improve our paper. Thank you so much for increasing your score; we are greatly encouraged by the recognition of our efforts.

---

### Author Response · Authors · 2023-08-20
**General Response**

We appreciate all the reviewers’ insightful comments and constructive feedback. We are encouraged that our constructed dataset LoRA is: (1) A novel (G9pJ, JcBn, 3Fyb, pVrF), expandable (G9pJ, 3Fyb), and challenging (G9pJ, 3Fyb), large-scale (3Fyb) VQA contribution; (2) Addressing diverse logical difficulties and furthering logical reasoning in VQA (G9pJ , JcBn, 3Fyb, KjfL, pVrF); (3) Equipped with automated scripts from knowledge extraction to image generation (G9pJ, JcBn, 3Fyb, KjfL, pVrF), and offering substantial advantage in customising the dataset for researchers (3Fyb, KjfL, pVrF); (4) Comprehensive in evaluating existing VQA and large-scale visual language models across various logical layers ( G9pJ , JcBn, 3Fyb, pVrF), and (5) highlighting the dataset’s potential as a future benchmark (pVrF) and promising avenues for future research (3Fyb).

We have already revised our paper and supplementary material to address reviewers’ suggestions. We addressed the following concerns.

**(1) Diversity of the domain knowledge**
(Reviewers 3Fyb, pVrF have noted that the LoRA dataset is primarily focused on food-and-kitchen scenarios and along with Reviewer G9pj, have suggested clarifying its broader applicability to other domains.)

The food and kitchen domains serve as representative scenarios to demonstrate the intricacies of logical reasoning in VQA.

Our approach and pipeline to create the dataset is generalizable. The framework utilizes the industry-standard OWL format for ontologies which is adaptable. It can work with any ontology (knowledge base) that adheres to the OWL specifications (standard W3C Web Ontology Language). (refer to Section 3.3.1, lines 165-168). We have provided additional details on the dataset generation processes, highlighting our method's applicability to a wide range of ontologies. (as detailed in Section C of the revised Supplementary, Dataset Generation Additional Details).

Logical constructs employed by humans, such as conjunction, disjunction, negation, and conditionals, are consistent across diverse knowledge domains, such as food-kitchen, outdoor scenarios, or mathematics. The essence of logical inference remains unchanged irrespective of the content domain. Our dataset seeks to probe the performance of current VQA methods across varying logical complexities, rather than their capacity to leverage external knowledge.

Regarding 3rd party tools: For LoRA's image generation, we used Blender, an open-source software, due to its rich repository of realistic food-and-kitchen objects. It also provides objects from other domains, allowing for expansion. Emerging AI tools like stable diffusion promise broader image diversity. Our future work intends to incorporate these advanced generative tools to enhance our dataset further.

**(2) Distinguishing Logical Reasoning from Knowledge-based and Visual Reasoning in VQA**
(Reviewer JcBn, G9pJ have asked about the relative prominence of visual reasoning and knowledge-based logical reasoning.)

Logical reasoning is different from knowledge-based reasoning, which queries the external knowledge base to provide a direct answer. It is also different from visual reasoning, which provides answers based on spatial, temporal, or visual relationships. Logical reasoning, however, deduces answers through elaborate multi-step chains of reasoning and involves navigating complex logical constructs.

LoRA’s foundational ethos is its emphasis on complex logical reasoning that requires multiple inference steps. This differentiates LoRA from other VQA datasets, such as knowledge-based reasoning or visual reasoning datasets. The dataset aims to pinpoint the logical complexity threshold that VQA models can address.

**Comparison Examples (Knowledge-based reasoning, Visual reasoning vs. Logical Reasoning):**

We thank the reviewers for referencing [1] and [2]. Questions like ''What type of architecture is shown in these buildings?'' or ''How many people will dine at this table?'' from these references mainly rely on external knowledge or visual cues.

From QLEVR [3], queries like ''Are all the cyan metallic triangular prisms on the brown plane?'' focus on attribute recognition, spatial relationships, and basic logical constructs. However, they lack advanced logical depth like conditionals.

In contrast, LoRA challenges further with: ''**If** we **don't** have milk, is there another **dairy** product, **not** necessarily fatty **but** protein-rich, suitable for breakfast?'' demands a deeper logical dive. This involves conditionals, negations, hierarchy, and existential constraints. ''Is the item to the left of the meat unsuitable for vegans and free of fat?''. These aren't merely visual or commonsense reasoning but entail intricate logical deliberations.

**Richness of Visual Content**: Moreover, while QLEVR is restricted to geometric imagery, LoRA uses diverse, realistic kitchen and food visuals, closely mirroring real-world scenarios.

---

### Decision · Program_Chairs · 2023-09-22

**Decision:**

Accept (Poster)

**Comment:**

Most of the reviewers value the proposed LoRA dataset as a novel, expandable, challenging, large-scale dataset. And the proposed LoRA dataset addresses the diverse logical difficulties and furthering logical reasoning in VQA, by equipping with automated scripts from knowledge extraction to image generation and offering substantial advantage in customising the dataset for researchers. Also some reviewers increases their ratings after the rebuttal, which somewhat clarify the reviewers' cocerns on the experimental results, reproducibility, domain specificity and expandability.

However, some comments are still not being well addressed, such as more detailed differentiation from the existing datasets (CLEVR, PTR, QLEVR), the visual reasoning in LoRA, and etc. The authors are highly suggested to incorperate these comments into their final revisions.